# Rejuvenating aged osteoprogenitors for bone repair

Joshua Reeves[1,2], Pierre Tournier[1†], Pierre Becquart[1†], Robert Carton[2], Yin Tang[3,4], Alessandra Vigilante[2], Dong Fang[3,4], Shukry J Habib[1]*

[1]Department of Biomedical Sciences, University of Lausanne, Lausanne, Switzerland; [2]Centre for Gene Therapy and Regenerative Medicine King's College London, London, United Kingdom; [3]Zhejiang Provincial Key Laboratory for Cancer Molecular Cell Biology, Life Sciences Institute Zhejiang University, Zhejiang, China; [4]Department of Ministry of Education, The Second Affiliated Hospital, Zhejiang University School of Medicine, Zhejiang, China

## eLife Assessment

Aging reduces tissue regeneration capacity, posing challenges for an aging population. In this **fundamental** study, Reeves et al. show that by combining Wnt-mediated osteoprogenitor expansion (using a special bandage) with intermittent fasting, calvarial bone healing can be restored in aged animals. Intermittent fasting improves osteoprogenitor function by rescuing aging-related mitochondrial dysfunction, which can also be achieved by nicotinamide mononucleotide (NMN) supplementation or by modulating the gut microbiome. By employing rigorous histological, transcriptomic, and imaging analyses in a clinically relevant model, the authors provide **compelling** evidence supporting the conclusions. The therapeutic approach presented in this study shows promise for rejuvenating tissue repair, not only in bones but potentially across other tissues.

*For correspondence:
shukryjames.habib@unil.ch

†These authors contributed equally to this work

**Abstract** Aging is marked by a decline in tissue regeneration, posing significant challenges to an increasingly older population. Here, we investigate age-related impairments in calvarial bone healing and introduce a novel two-part rejuvenation strategy to restore youthful repair. We demonstrate that aging negatively impacts the calvarial bone structure and its osteogenic tissues, diminishing osteoprogenitor number and function and severely impairing bone formation. Notably, increasing osteogenic cell numbers locally fails to rescue repair in aged mice, identifying the presence of intrinsic cellular deficits. Our strategy combines Wnt-mediated osteoprogenitor expansion with intermittent fasting, which leads to a striking restoration of youthful levels of bone healing. We find that intermittent fasting improves osteoprogenitor function, benefits that can be recapitulated by modulating NAD$^+$-dependent pathways or the gut microbiota, underscoring the multifaceted nature of this intervention. Mechanistically, we identify mitochondrial dysfunction as a key component in age-related decline in osteoprogenitor function and show that both cyclical nutrient deprivation and Nicotinamide mononucleotide rejuvenate mitochondrial health, enhancing osteogenesis. These findings offer a promising therapeutic avenue for restoring youthful bone repair in aged individuals, with potential implications for rejuvenating other tissues.

## Introduction

Flat bones like the sternum, scapula, pelvic bones, and skull bones play crucial roles in the body, including protecting organs and anchoring muscles. Their thin structure makes them prone to fractures (*Huelke and Compton, 1983*), leading to pain, disability, and complications like internal bleeding or

organ damage (*Ren et al., 2020*). As individuals age, flat bones become more brittle, increasing the risk of fractures (*Beedham et al., 2019*). Skull fractures, common in head injuries at any age (*Carson, 2009*), carry a higher mortality rate even when considering trauma severity (*Tsai et al., 2022*). Aging affects the shape and fragility of calvarial bones (*Cotofana et al., 2018*; *Torimitsu et al., 2014*), but the impact on osteogenic tissues supporting bone repair, such as the calvarial periosteum and suture mesenchyme remains elusive.

In the calvaria, bones form and repair through intramembranous ossification (*van den Bos et al., 2008*). Osteoblasts which deposit pre-mineralized bone tissue (*Bianco and Robey, 2015*) arise from CD90[+] osteoprogenitor populations (*Chan et al., 2015*; *Chan et al., 2013*; *Kim et al., 2016*), which are derived from heterogeneous populations of skeletal stem cells (SSCs; *Chan et al., 2015*; *Chan et al., 2013*; *Kim et al., 2016*). Osteoprogenitors reside in soft tissues alongside endothelial and other cell types that interface with mineralized bone tissue. The two main described osteoprogenitor niches are the periosteum, which overlays all flat bones (*Debnath et al., 2018*), and the suture mesenchyme, only present in the skull, between calvarial bones (*Wilk et al., 2017*). Localized Wnt ligands (Wnts) play crucial roles in regulating osteogenesis, influencing stem cells, progenitors, and mature osteoblasts (*Garcin and Habib, 2017*; *Liu et al., 2009*; *Lowndes et al., 2016*; *Matsushita et al., 2020*; *Okuchi et al., 2021*). We recently showed that the Wnt3a-bandage, a transplantable polymer that harbors covalently immobilized Wnt3a proteins, combined with an engineered human osteogenic tissue, can promote bone repair in adult immunodeficient mice (*Okuchi et al., 2021*).

In this study, using a clinically relevant immunocompetent mouse model, we examined how aspects of the periosteum and suture mesenchyme change with age, how they respond to injury, and the influence of localized Wnts throughout aging.

Young mice achieve almost complete bone repair of calvarial defects considered 'critical sized' in adults, which have limited self-healing ability. We found that age-related changes in bone tissue begin to manifest during adulthood, including increased bone porosity, altered cellular mechanical cues, and declines in mitochondrial activity, vascularity, and the proportion and function of CD90[+] osteoprogenitor cells within the periosteum and suture mesenchyme. While Wnt bandage increases CD90[+] cells at the defect site regardless of age, this increase only sufficiently promotes repair comparable to young animals in adult mice, not aged mice. Our study characterized a more pronounced decline in osteogenic compartments in aged mice, encompassing reduced CD90[+] cell proportion, function, and mitochondrial bioenergetics, alongside other contributing factors. This multifaceted decline leads to severe age-related bone repair deficiency, only partially alleviated by the Wnt3a-bandage, underscoring the need for combinatorial approaches.

To counteract age-related decline in bone repair, we combined localized Wnt3a stimulation with systemic metabolic interventions. In aged mice with critical-size defects, alternating-day intermittent fasting restored osteogenic compartment health, improving CD90[+] cell function as well as bone remodelling and significantly enhanced bone repair to levels comparable to younger animals. This effect was recapitulated by even shorter-term supplementation of the gut bacteria *Akkermansia muciniphila* or nicotinamide mononucleotide (NMN – an NAD[+] precursor) in aged mice, highlighting the multifaceted mechanisms of action of intermittent fasting. We further identify mitochondrial dysfunction as a key component in age-related decline in osteoprogenitor function and show that both cyclical nutrient deprivation or NMN rejuvenate mitochondrial health and enhance osteogenesis in vitro.

In summary, our findings highlight that aging tissues retain the potential for health improvements through interventions. Our data reveals an intriguing avenue of research where metabolic and microbiome interventions can be repurposed to treat tissue damage in aged repair-deficient individuals.

## Results
### Parietal bone structure deteriorates with aging

Initially, we analyzed the changes that occur in calvarial bone of female C57BL/6J mice during aging at the macro-structure. Female mice were selected for this study due to their distinct characteristics, such as delayed healing and higher fracture risk (*Haffner-Luntzer et al., 2021*; *Ortona et al., 2023*), which provide a more challenging model for evaluating bone repair strategies. Young mice (6 weeks) are within the period of fastest bone formation and growth, Adult mice (13 weeks) at healthy homeostasis, and Aged mice (>88 weeks) correspond to declining skeletal health as a model of >70-year-old

humans (*Dutta and Sengupta, 2016*). We evaluated the calvaria of these animals using micro-computed tomography (microCT). The proportion of the total bone volume taken up by mineralized bone tissue (Bone Volume/Total Volume, BV/TV) significantly decreased with age (*Figure 1a–b*). Cortical bone mineral density (Ct.BMD), the measure of the density of the mineralized bone tissue alone, showed that Aged mice have similar mineral density to Young and Adult mice (nonsignificant differences). The overall thickness from outer surface to inner surface of the parietal bone significantly increased with age (*Figure 1—figure supplement 1a-d*).

As parietal bone formation and maintenance is regulated by the cells within the periosteum and suture mesenchyme (*Debnath et al., 2018*; *Li et al., 2021*; *Maruyama et al., 2016*), we next defined the changes that occur within these compartments during aging.

## Age-related changes in vascularization, nuclear morphology, F-actin levels and osteoprogenitor proportions in osteogenic compartments

Blood vessels in the calvaria bone support tissue survival by delivering oxygen and nutrients, regulating bone metabolism, and contributing to its maintenance and remodeling, while tissue stiffness influences cell behavior and osteoprogenitors differentiate into cells essential for these processes. Therefore, we aimed to characterize blood vessels, nuclear and cytoskeletal morphology, and osteoprogenitor proportions within the periosteum and suture mesenchyme. Initially, we established that 97% of endothelial cells in the periosteum and the suture mesenchyme (osteogenic compartments) express Endomucin and CD31 (*Figure 1—figure supplement 1e, f*). We found that the proportion of Endomucin-positive (Emcn+) blood vessels in the periosteum significantly decreases with age (*Figure 1c–d*, *Figure 1—figure supplement 1g-i*). Further vascular histomorphometry shows that, in the periosteum, blood vessel lumen area is consistent throughout aging, indicating that it is the decrease in the blood vessel number per area of tissue that drives the declining blood vessel density. In addition, we find that the thickness of the periosteum itself also decreases rapidly with age (*Figure 1—figure supplement 1k, l*). Taking increased periosteum thickness and density of blood vessels into account, the estimated blood supply (vessel density multiplied by periosteum thickness) of the whole Young periosteum is over 12-fold higher than that of Aged periosteum. In the suture mesenchyme, blood vessels are larger than in the periosteum, and overall, the vascularization remains stable throughout aging (*Figure 1—figure supplement 2a, b*).

Vascular phenotypes can be influenced by tissue mechanics (*Xie et al., 2018*). Therefore, we aimed to measure mechanically driven aspects of cellular morphology in periosteal and suture mesenchymal cells during aging. Nuclear shape is a product of intracellular (*Dahl et al., 2008*) and extracellular factors, such as the stiffness of the surrounding matrix (*Lovett et al., 2013*). Morphologically and confirmed by the mean Nuclear Aspect Ratio (NAR, long axis length divided by short axis length) we found significant elongation of nuclei in Aged mice periosteum and to a lesser extent, in the suture mesenchyme (*Figure 1e–f*). Lamin B1, a key nuclear lamina protein involved in regulating nuclear function and organization, showed no significant changes across age groups (*Bin Imtiaz et al., 2021*; *Kim, 2023*; *Tran et al., 2016 Figure 1—figure supplement 2i–j*).

Nuclear morphology is frequently interlinked with actin cytoskeleton organization, influencing cell migration, proliferation (*Ambriz et al., 2018*) and the response to mechanical force imparted on them through the extracellular matrix (*Fletcher and Mullins, 2010*; *Stricker et al., 2010*). We quantified filamentous actin (F-actin) using fluorescent-Phalloidin staining in periosteum and suture mesenchyme (*Figure 1g–h*, *Figure 1—figure supplement 2h*). Aging led to a progressive increase of F-actin, peaking at a 2.5-fold rise between Young and Aged mice in suture mesenchyme.

Beyond morphological and cytoskeletal changes, we questioned whether aging periosteal cells experience changes to their cellular energy supply. Adenosine triphosphate (ATP) generation, crucial for cell function, relies largely on mitochondrial membrane potential (*Zorova et al., 2018*). This can be measured by determining the intensity of Tetramethylrhodamine, methyl ester (TMRM), which accumulates in active mitochondria with intact membrane potentials in live cells (*Creed and McKenzie, 2019*). Ex vivo TMRM intensity was significantly higher in freshly harvested samples of Young mouse periosteum, than in Adult and Aged mice (*Figure 1i–j*). This finding is unlikely to be an artefact of differential ECM densities interfering with diffusion as performing the same experiment in explanted cells (i.e. without ECM, *Figure 1—figure supplement 2c, d*) gave a similar trend of results. We also

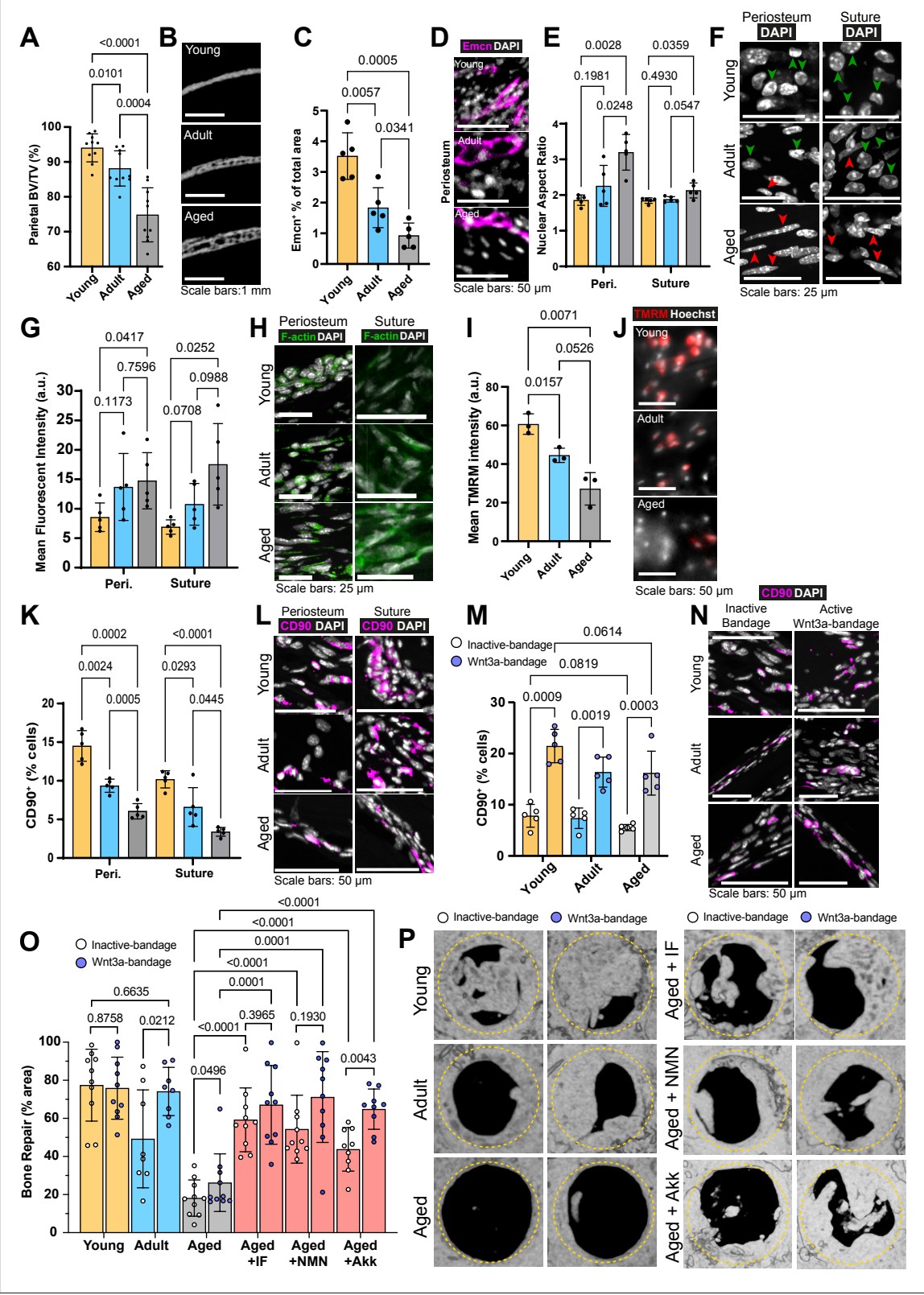

**Figure 1.** Changes to the calvarial bone, periosteum and suture mesenchyme, and bone repair with aging and interventions. (**A**) Parietal Bone Volume/Total Volume (BV/TV) and (**B**) representative transverse micro computed tomography (microCT) sections in Young, Adult, and Aged mice. Statistical analysis: Welch's test in n=10 mice per condition. (**C**) Mean cross-sectional area taken up by Endomucin (Emcn)-positive vascular cells and (**D**) representative immunofluorescence staining of Emcn (magenta) and DAPI (grey) in Young, Adult, and Aged mouse periosteum. Statistical analysis:

*Figure 1 continued on next page*

*Figure 1 continued*

Welch's tests in n=5 mice per condition. (**E**) Nuclear aspect ratio (long axis divided by short axis) of cells within the periosteum and suture mesenchyme and (**F**) representative DAPI-stained nuclei in Young, Adult, and Aged mice. Statistical analysis: Welch's tests in n=5 mice per condition. Green and red arrows highlight rounded and elongated nuclei, respectively. (**G**) Mean Phalloidin-488 fluorescence intensity and (**H**) representative images for filamentous actin (F-actin, green) and DAPI (grey) in Young, Adult, and Aged mouse periosteum and suture mesenchyme. Statistical analysis: Welch's tests in n=5 mice per condition. (**I**) Mean intensity and (**J**) representative images of live TMRM staining (red) and Hoechst (grey) in Young, Adult, and Aged mouse periosteum ex vivo. Statistical analysis: Welch's tests in tissue from n=3 mice per condition. (**K**) Percentage of CD90$^+$ cells within the periosteum and suture mesenchyme, expressed relative to the total cellular number (DAPI$^+$) in Young, Adult, and Aged mice. Representative images (**L**) show CD90 immunostaining (magenta) and DAPI (grey). Statistical analysis: Welch's tests in n=5 mice per condition. (**M**) Percentage of CD90$^+$ cells within calvarial defects treated with Wnt3a-bandages (blue points) or inactive controls (white points), expressed relative to the total cellular number (DAPI$^+$) in Young, Adult, and Aged mice. (**N**) Representative images show CD90 immunostaining (magenta) and DAPI (grey). Statistical analysis: ratio-paired two-tailed t tests in n=5 mice per condition. (**O**) Percentage of repair and (**P**) representative top-down microCT images within calvarial defects treated with Wnt3a-bandages (blue points) or inactive controls (white points). Statistical analysis within age groups: ratio-paired two-tailed t tests; between age groups and Aged mice following the IF protocol or NMN or Akk supplementation: Welch's tests; Young, Aged, Aged + IF, Aged + NMN, n=10 mice; Aged + Akk, n=9; Adult, n=8. Complete statistical analysis between aged groups and treatments can be found in *Supplementary file 1*. The yellow dashed circle highlights the 2 mm diameter defect size.

The online version of this article includes the following figure supplement(s) for figure 1:

**Figure supplement 1.** Bone morphometry, histology, and vasculature changes during aging and post-interventions.

**Figure supplement 2.** Quantification of Emcn+ vasculature, F-actin, Lamin B1, and mitochondrial membrane potential in osteoprogenitor compartments during aging and post-interventions.

**Figure supplement 3.** Characterization of bandage activity and histology of newly formed bone under Wnt bandage during aging and post-interventions.

**Figure supplement 4.** Characterization of Ki67, β-catenin, and MMP2 expression in CD90+ Cells in adult and aged animals.

confirmed equal Hoechst diffusion and staining throughout the ex vivo tissue experiment (*Figure 1—figure supplement 2e, f*).

Given these age-related differences in the osteogenic compartments, we aimed to elucidate changes to the CD90$^+$ osteoprogenitors (*Chan et al., 2015*; *Kim et al., 2016*; *Figure 1—figure supplement 2g*). Using in situ immunofluorescence, we found that the proportion of CD90$^+$ in the periosteum and the suture mesenchyme significantly decreases with age, with only around one third retained in Aged mice in each compartment (*Figure 1k–l*).

These tissue and cell-level changes indicate a significant and multifaceted age-related decline in the calvarial osteogenic compartments. The periosteum and the suture mesenchyme also serve as the primary sources of cells that contribute to osteogenesis during calvarial bone repair (*Jeffery et al., 2022*; *Wilk et al., 2017*). Hence, we next characterized how calvarial bone repair is affected by age and how the nearby cells respond to the Wnt3a-bandage.

## Wnt3a-bandages increase CD90$^+$ cells in repair areas of Aged and Adult mice and enhance bone repair

Critical-sized calvarial bone defects, challenging to heal without intervention, serve as models to quantitatively evaluate bone repair. Our prior work demonstrated that applying a Wnt3a-bandage and cellular osteogenic construct to the defect site significantly enhances bone repair in adult immuno-compromised mice (*Okuchi et al., 2021*). Considering the crucial role of the immune system in bone healing (*Loi et al., 2016*) and the prevalence of bone fractures in immunocompetent individuals, we assessed bone repair in female wildtype (C57BL/6 J) mice of varying ages. Mice underwent bilateral critical-sized (2 mm) full-thickness calvarial defect surgery in the parietal bones and were treated with one active Wnt3a-bandage and one chemically inactivated Wnt3a control bandage. Wnt3a-bandage activity was confirmed in vitro (*Figure 1—figure supplement 3a–c*) before transplantation.

We assessed bone repair by measuring the area percentage of re-mineralized tissue within the defect site using microCT after 10 weeks. Interestingly, Young mice exhibited robust bone healing even without intervention, highlighting the natural regenerative capacity of bone in young mice. (*Figure 1o–p*, *Supplementary file 1*). In adult mice, Wnt3a-bandage treatment significantly enhanced bone repair compared to the control bandage, achieving levels comparable to those observed in young mice. Aged mice exhibited severely impaired repair with both bandages, though Wnt3a-bandage showed a modest but significant increase. New bone formed under Wnt3a-bandage

displayed improved lamellar structure (striations) akin to mature undamaged bone (*Figure 1—figure supplement 3d*).

To uncover cellular mechanisms behind the improved healing, we examined CD90+ osteoprogenitor cells in the defect site. The proportion of CD90+ cells was significantly increased by the Wnt3a-bandage across age groups, compared to control bandages (*Figure 1m–n*). Wnt3a-bandages locally increase CD90+ cells, but despite similar numbers across ages, the effect of the Wnt3a-bandage on bone repair in Aged mice was limited. We hypothesized, therefore, that the functionality of CD90+ cells, rather than their number alone, may drive age-related differences in bone repair outcomes.

To investigate this, we initially examined the influence of the Wnt-bandage on CD90+ cells across various age groups in terms of Wnt/β-catenin signaling. The proportion of nuclear β-catenin (*Figure 1—figure supplement 4a–c*), indicative of Wnt/β-catenin signaling activation (*Archbold et al., 2012*), was significantly increased in CD90+ cells near the active Wnt3a-bandage compared to the inactive bandage. Importantly, the proportion of nuclear β-catenin remained similar regardless of age, suggesting the Aged CD90+ have the capacity to activate the pathway similar to younger CD90+ cells.

We then explored the effects of the Wnt3a-bandage on CD90+ cell proliferation and migration near the bandage. Our results demonstrate that the proportion of CD90+ stained with the proliferation marker Ki67 or MMP2, a Type IV collagenase enzyme linked to migration (*Ries et al., 2007*) and ECM remodeling in flat bones (*Egeblad et al., 2007*; *Mosig et al., 2007*; *Zhong-Sheng et al., 2022*), appeared increased in CD90+ near Wnt3a-bandage. While this increase was not statistically significant compared to inactive control bandage groups in Aged mice, it was statistically significant in Adult mice (*Figure 1—figure supplement 4b–e*).

Considering that CD90+ cells inside the defect are ultimately derived from cells that are harbored in the nearby periosteum and suture mesenchyme, we next analyzed the transcriptomes of CD90+ cells to glean insights into their functional differences during aging.

## Transcriptional shifts occur in calvarial CD90+ cells with aging

Single-cell RNA sequencing (scRNAseq) was performed on live cells immediately following dissection and tissue digestion from the periosteum and suture mesenchyme in Young, Adult, and Aged mice, with data processing to deplete immune (CD45+ (*Ptprc*)) cells. CD90+ (*Thy1*) subpopulations of cells (cluster 0, 1 and 3 in the periosteum; cluster 10 in the suture mesenchyme) were identified within both tissues (*Figure 2—figure supplement 1a–h*), with high transcriptomic similarity to cells expressing *Prrx1*, Cathepsin K (*Ctsk*) and *Ddr2* (*Bok et al., 2023*; *Debnath et al., 2018*; *Figure 2—figure supplement 2*), markers of periosteal and suture SSCs (*Duchamp de Lageneste et al., 2018*; *Wilk et al., 2017*). We also found that the CD90+ population includes minor subsets of cells that express other SSC/progenitor markers including *Cd200/Itgav* (*Menon et al., 2021*), *Gli1* (*Zhao et al., 2015*), *Axin2* (*Maruyama et al., 2016*), and α-SMA (*Acta2*; *Ortinau et al., 2019*). Therefore, based on functional studies (*Kim et al., 2016*; *Nusspaumer et al., 2017*) and our transcriptomic analysis, the CD90+ population encompasses a heterogeneous group of cells that are capable of osteogenic differentiation. Next, we will describe selected transcriptomic changes in CD90+ cells during aging. For Gene ontology number, comparisons and statistics, please see *Supplementary file 2*.

Comparing Young CD90+ cells with Aged CD90+ cells in the periosteum (*Figure 2—figure supplement 3a*), 542 genes were significantly upregulated. Of these, 38 corresponded to Gene Ontology (GO) term 'Cytosolic ribosome', linked to increased 'Collagen-containing extracellular matrix'. Similarly in Young vs Aged suture CD90+ cells (*Figure 2—figure supplement 3b*), 879 significantly upregulated genes are enriched for 'Structural constituent of ribosome' and 'Extracellular matrix structural constituent'. Young mice at 6 weeks of age undergo significant skeletal growth (*Jilka, 2013*), hence this data closely corroborates the increase in collagenous matrix deposition (both soft and bone tissue) observed in histological sections of bone defects in Young mice.

Given the effect of the Wnt3a-bandage on the Aged bone repair environment, we next probed the differentially expressed genes for components of the Wnt/β-catenin pathway. The transcript's expression of Wnt2 ligand, often implicated in the Wnt/β-catenin pathway (*Goss et al., 2009*; *Jung et al., 2015*), was significantly downregulated with age in suture CD90+ cells (*Figure 2—figure supplement 3b d*). Further, Wnt co-receptor *Lrp6* expression significantly declined with age in periosteal CD90+

cells (*Figure 2—figure supplement 3a*). Collectively, this suggests compromised Wnt signaling in the periosteum and suture mesenchyme during aging.

At the Adult and Aged timepoints, bone is at healthy and declining homeostasis, respectively. We compared Adult and Aged CD90$^+$ cells in the periosteum (*Figure 2—figure supplement 3c*). Here, the 637 genes significantly upregulated in Adult vs Aged were vastly enriched for GO terms 'NADH dehydrogenase complex assembly', 'NADH dehydrogenase (ubiquinone) activity' and 'Oxidative phosphorylation'. Likewise, in the suture mesenchyme (*Figure 2—figure supplement 3d*), the 610 genes significantly upregulated in Adult vs Aged CD90$^+$ cells were also enriched for 'Mitochondrial ATP synthesis coupled proton transport' and 'Oxidative phosphorylation'. Using the UCell method (*Andreatta and Carmona, 2021*), we found significantly reduced Complex I gene scoring with age in the suture mesenchyme and the periosteal CD90$^+$ cells (*Figure 2—figure supplement 3e–f*). These results, coupled with live measurements of mitochondrial membrane potential (*Figure 1i–j*), indicate an age-related decline in mitochondrial metabolism and electron transport chain (ETC) complexes within CD90$^+$ cells of the whole periosteum and suture mesenchyme. Importantly, we did not observe distinct changes in the signature of Glycolysis or lipid metabolism across age groups in CD90$^+$ cells in the osteogenic compartments (*Figure 2—figure supplement 3j–m*).

Analysis with the CellChat package (*Jin et al., 2021*) shows inferred signaling heterogeneity within the periosteal CD90$^+$ population (*Figure 2—figure supplement 4a*). Cells within the smaller CD90$^+$ cluster (cluster 3, *Figure 2—figure supplement 1a–d*) receive incoming Wnt-signals and also Periostin-signalling, which upregulates Wnt/β-catenin pathway activity in osteogenesis (*Lv et al., 2015*; *Zhang et al., 2017*). This cluster is significantly upregulated for markers of osteogenic differentiation such as *Runx2*, *Mme* (Osteonectin/CD10) (*Ding et al., 2020*; *Granéli et al., 2014*), *Bgn* (*Ye et al., 2012*), *Mgp* (*Zhang et al., 2019*), *Gas6* (*Shiozawa et al., 2010*), and *Igbp7* (*Zhang et al., 2018*), and is hereby termed the P1 CD90$^+$ population. Cells within the larger cluster (clusters 0 and 1) both send and receive Periostin and Wnt-signals, and are significantly upregulated for markers associated with stemness, such as *Klf4* and *Aldh1a3* (*Wanandi et al., 2018*), and negative regulators of osteogenic differentiation such as *Bmp3* (*Kokabu et al., 2012*), and *Ccn3* (*Matsushita et al., 2013*). Hereby these cells were termed P2 CD90$^+$.

The periosteal P2 CD90$^+$ cell population shifts with age between two categories, named '*Igf1*$^{high}$' and '*Igf1*$^{low}$' in relation to their differential expression of *Igf1* (*Figure 2a–c*). *Igf1* is linked to bone anabolism through its dual roles in enhancing Wnt-signaling and exerting metabolic effects (*Locatelli and Bianchi, 2014*). Young P2 CD90$^+$ cells were restricted to the *Igf1*$^{high}$ grouping (92.0%), while Aged CD90$^+$ cells were predominantly in the *Igf1*$^{low}$ grouping (66.3%; *Figure 2a–b*). Adult CD90$^+$ cells were more evenly split (56.8% *Igf1*$^{high}$, 43.2% *Igf1*$^{low}$). *Igf1*$^{low}$ cells were significantly upregulated for 85 genes (*Figure 2c*), linked to 'Cell motility', driven by genes such as *Mmp14* (*Almalki and Agrawal, 2016*) and *Itgb7* (*Novoseletskaya et al., 2020*). *Igf1*$^{high}$ cells were significantly upregulated for 204 genes linked to 'Collagen-containing extracellular matrix', 'Oxidative phosphorylation' and 'Respiratory chain complex'. The enrichment is driven by significantly upregulated genes such as NADH dehydrogenase genes (*mt-Nd1* and *mt-Nd2*), *Col1a1*, *Sparc*.

Likewise, in the suture mesenchyme, analysis of inferred cellular state and function (*Figure 2d–e*) revealed a distinct subpopulation of CD90$^+$ cells. This group of cells was significantly upregulated for 855 genes linked to 'Mitotic cell cycle' and 'ATP-dependent activity'. In Young mice, this population, termed 'Proliferating' cells, represented 48.72% of the CD90$^+$ population, whereas it markedly declined to around 8% in both Adult and Aged mice (*Figure 2d–e*).

When comparing CD90$^+$ cell populations between every two age stages, several genes are conserved in all three comparisons (*Figure 2—figure supplement 4b, c*), including: *Cavin1* (*Volonte and Galbiati, 2020*), *S100a4* (*Wagner et al., 2009*), *Vcan* (*Li et al., 2019a*), *Mmp14* (*Haage et al., 2014*), and *Nsg1* (*Wiese et al., 2019*). Each of these genes has been identified as being involved in senescence or the cellular response to aging tissues. Likewise, in the suture CD90$^+$ cells, conserved aging-related genes included senescence-related genes *Pim1*(*Yang et al., 2017*), *Pdcd4* (*Kang et al., 2002*); and the senescence-associated secretory profile (SASP) gene *Ccl6* (*Gillispie et al., 2021*). Further comprehensive senescence signature confirms the significant increase associated with age in CD90$^+$ cells (*Figure 2—figure supplement 4d*). Also conserved were: pro-bone resorption *Mmp9* (*Zhu et al., 2020*), and adipogenesis-related genes *ApoD* (*Provost et al., 1991*) and *Slc16a3* (MCT4; *Petersen et al., 2017*).

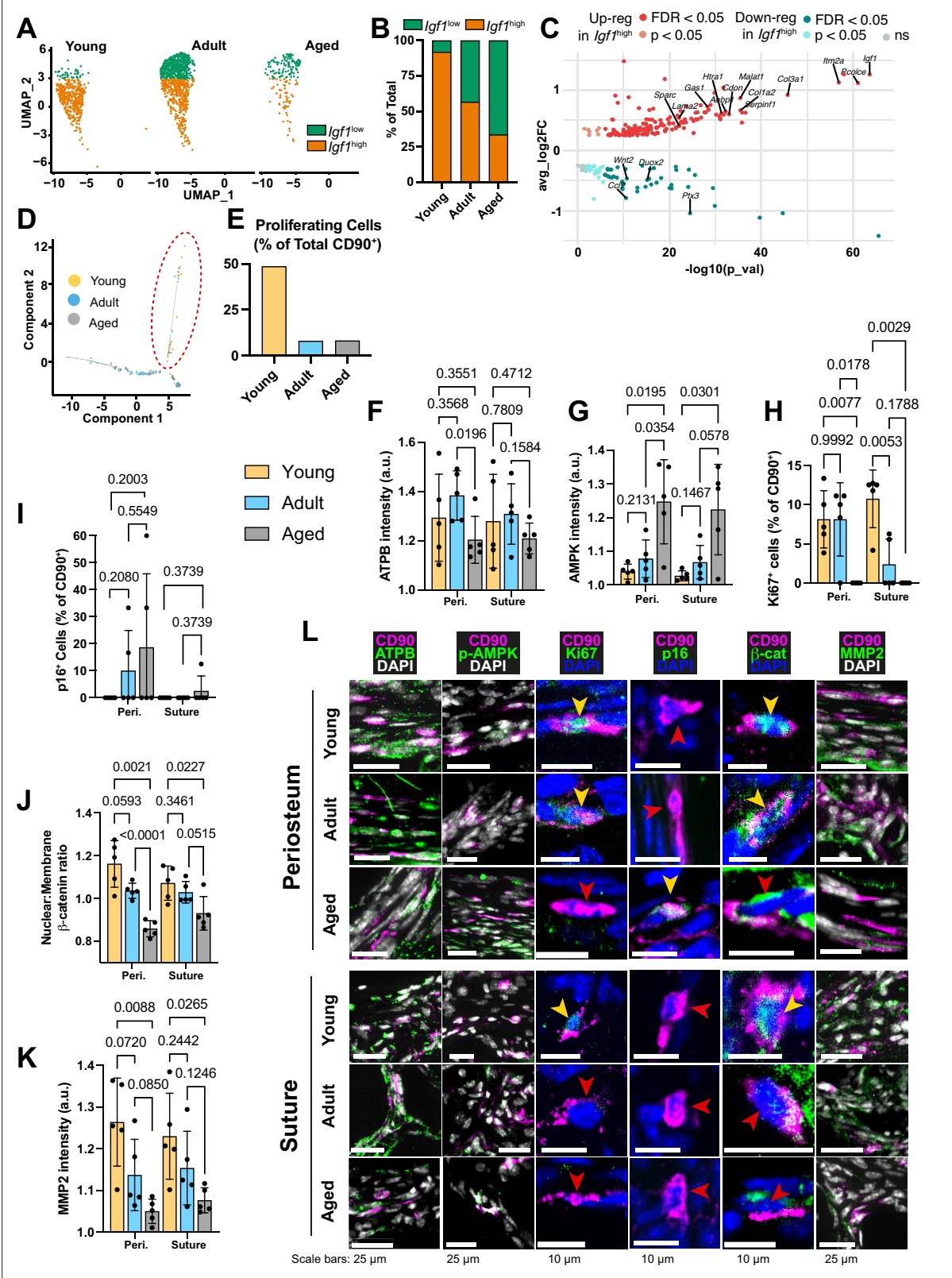

**Figure 2.** Transcriptomic and protein expression characterization of CD90+ cells during aging. (**A**) UMAP-plot showing the distribution of a subpopulation of CD90+ periosteal cells with aging, categorized as *Igf1*high (orange) and *Igf1*low (green). (**B**) Percentage of the total CD90+ subpopulation categorized as either *Igf1*high or *Igf1*low. Young: *Igf1*high (n=425 cells), *Igf1*low (n=37); Adult: *Igf1*high (n=553), *Igf1*low (n=421); Aged: *Igf1*high (n=61), *Igf1*low (n=120). (**C**) Volcano plot showing differentially-expressed genes in the comparison of all *Igf1*high cells (n=1039) against all *Igf1*low cells (n=578). Statistical

*Figure 2 continued on next page*

*Figure 2 continued*

analysis: Wilcoxon Rank Sum test via the FindMarkers function in the Seurat R package; upregulated (pink: p<0.05; red: False discovery rate (FDR) -adjusted p<0.05), downregulated (cyan: p<0.05; teal: FDR-adjusted p<0.05), and non-significant (ns) genes are displayed on axes of log fold expression (*Igf1*high over *Igf1*low) (y, avg_log2FC) against p transformed as its negative log value. (**D**) Dimensional reduction plot (Monocle package *Trapnell et al., 2014*) of suture mesenchyme CD90+ populations in Young (yellow), Adult (blue), and Aged (grey) mice. (**E**) Percentage of CD90+ categorized as 'proliferating' (inside red dashed ellipse) vs 'non-proliferating' (outside ellipse). Young: proliferating (n=19 cells), nonproliferating (n=20); Adult: proliferating (n=5), non-proliferating (n=57); Aged: proliferating (n=5), nonproliferating (n=56). (**F**) Normalized intensity staining for ATPB in CD90+ cells. (**G**) Normalized intensity staining for phospho-AMPK in CD90+ cells. (**H**) Proportion of CD90+ cells that are Ki67+. (**I**) Proportion of CD90+ cells that are p16+. (**J**) Ratio of Nuclear:Membrane β-catenin staining in CD90+ cells. (**K**) Normalized intensity staining for MMP2 in CD90+ cells. For all plots (**F–K**), Young (yellow), Adult (blue), Aged (grey), periosteum (3 left columns), suture mesenchyme (3 right columns), statistical testing: Welch's tests in n=5 mice per condition. (**L**) Representative images depicting CD90 (magenta), DAPI (grey/blue as indicated by column), and functional markers (green) as indicated by column. Yellow arrows indicate marker-high/positive cells; red arrows indicate marker-low/negative cells.

The online version of this article includes the following figure supplement(s) for figure 2:

**Figure supplement 1.** UMAP-coordinate plot of *Thy1* (CD90).

**Figure supplement 2.** UMAP-coordinate plot of the entire cell populations from Periosteum.

**Figure supplement 3.** Differential gene expression and UCell signature scoring in CD90+ osteoprogenitor populations across age and tissue compartments.

**Figure supplement 4.** Cell-Cell Signaling, Differential Gene Expression, and Overlap Analysis in CD90+ Osteoprogenitor Populations Across Age and Tissue Compartments.

## Aging calvarial CD90+ cells undergo a decline in energy supply, proliferation, Wnt/β-catenin, and ECM remodeling capacity

Next, we aimed to investigate the transcriptionally-identified decline in energy supply, proliferation, Wnt/β-catenin pathway, and migration in CD90+ cells at the protein level through immunofluorescent staining of marker proteins.

The catalytic beta subunit of ATP synthase (ATPB), an essential component of the mitochondrial ATP synthesis machinery, serves as a marker for assessing energy-related mitochondrial function (*Li et al., 2019b*). Cellular ATPB levels in suture mesenchyme CD90+ cells is similar across age groups, but in the periosteal CD90+ cells, ATPB level is lower in Aged mice compared to Adults (*Figure 2f and I*). Phosphorylated AMPK (phospho-AMPK), a sensor of ATP depletion (*Gowans and Hardie, 2014*), exhibits a negative correlation with energy availability. In both periosteum and suture CD90+ cells, mean phospho-AMPK was significantly higher in Aged mice (*Figure 2g and I*). Taken together, these findings suggest a potential deficiency in oxidative phosphorylation machinery in CD90+ cells with age.

Ki67 marks active proliferation, while cyclin-dependent kinase inhibitors p16 is involved in various cellular functions and is linked to senescence and aging in several tissues (*Safwan-Zaiter et al., 2022*; *Stein et al., 1999*). Calvarial CD90+ cells were assessed for nuclear expression of the two markers. Ki67+ CD90+ cells were observed in the periosteum of both Young and Adult mice but not in Aged mice. In the suture, the proportion of Ki67+ CD90+ was significantly higher in Young than in Adult mice with none detected in Aged mice (*Figure 2h and I*). The proportion of p16+ CD90+ cells appeared to increase with Age (*Figure 2i and I*). Together, nuclear p16 expression and a lack of Ki67 expression suggest that Aged CD90+ cells are not actively proliferating, corroborating inferred findings from transcriptomics.

Next, we assessed the level of nuclear β-catenin, an indicator of Wnt/β-catenin pathway activation. Aged CD90+ cells in both periosteum and suture mesenchyme showed significantly reduced nuclear localization of β-catenin compared to Young (*Figure 2j and I*). Only Adult CD90+ cells in the periosteum had significantly higher nuclear β-catenin when compared to Aged mice.

Finally, in both periosteum and suture mesenchyme, Aged CD90+ cells expressed the lowest amounts of MMP2 (*Figure 2k–I*). While MMP2 is a secreted protein, only intracellular protein was quantified, to enable the analysis to be restricted to MMP2 expressed by CD90+ cells alone. It corroborates the downregulation of genes associated with migration in the transcriptomic analysis.

Overall, our protein-level analysis largely aligns with the transcript-level findings, insinuating that age-related alterations in periosteum and suture mesenchyme, and consequently bone repair, could be considerably impacted by metabolic differences within the CD90-expressing population.

Given that increasing the number of CD90[+] cells inside the defect alone produced only a minor improvement to Aged bone repair, we developed a systemic approach to target cellular metabolism. We aimed to combine this systemic approach with the use of the Wnt3a-bandage to ultimately facilitate the supply of competent osteogenic cells for bone repair.

## Intermittent fasting improves bone health and bone repair in aged mice

Intermittent fasting (IF) can improve metabolic control by various measures (*Longo et al., 2021*) and bone mineral density in adult animals (*Veronese and Reginster, 2019*). We produced critical-sized calvarial defects in Aged mice, treated them with Wnt3a-bandages and inactive controls. These mice were then maintained on an alternating-day diet of self-limited feeding and complete fasting, hereby termed the intermittent fasting diet. Importantly, over the course of a week, mice consume the same amount of food during the IF diet as during the self-limited feeding controls (ad libitum, AL) diet (*Figure 3—figure supplement 1a–c*), and is therefore distinct from an overall reduction in calories (i.e. caloric restriction). Fasting blood glucose levels and the AUC from the intraperitoneal glucose tolerance test (IPGTT) were significantly lower in mice following an IF diet compared to those on an AL diet (*Figure 3—figure supplement 1d–g*). These results indicate improved insulin sensitivity, enhanced glucose metabolism, and better overall metabolic control (*Andrikopoulos et al., 2008*; *McGuinness et al., 2009*; *Nakrani et al., 2024*).

Remarkably, Aged mice subjected to the IF diet repaired 59.24% of the defect within 10 weeks under the inactive bandage, higher even than that of Adult mice +inactive bandage (*Figure 1o–p*). With the addition of the Wnt3a-bandage, bone repair area was increased to 67.10%. This combination approach of the local Wnt3a-bandage and post-fracture IF results in Aged calvarial bone repair rivaling that of Young mice. The IF diet also significantly increased the solid bone proportion (BV/TV; *Figure 3a–b*), making it similar to the much younger Adult bone.

## Intermittent fasting reverses aspects of cellular and tissue aging in the periosteum and suture mesenchyme

Compared to Aged mice, in Aged + IF, Emcn[+] vascularity in the periosteum was significantly increased (*Figure 3c–d*). Additionally, in the suture mesenchyme, actin cytoskeleton staining was markedly decreased (*Figure 3e–f*), along with a significant reduction in age-related nuclear elongation both in the suture and periosteum (*Figure 3g–h*). The IF diet did not impact Lamin B1 protein levels (*Figure 1—figure supplement 2i–j*). The IF diet appeared to elevate the proportion of CD90[+] cells in the periosteum, while in the suture it significantly increased compared to AL (*Figure 3i–j*).

To gain a deeper understanding of how IF affects Aged CD90[+] cells, we conducted further scRNAseq analysis. We maintained Aged mice on the alternating-day IF diet for 10 weeks before performing scRNAseq on the periosteum and suture mesenchyme (*Source data 2, supplementary tables 9,10,12*).

Comparing CD90[+] cells of Aged periosteal post IF diet with Aged AL diet controls revealed 98 significantly upregulated genes (*Figure 4a*). These were enriched mainly for 'Response to cytokine' and similar GO terms. Of the genes upregulated with the IF diet, 10 were individually statistically significant when adjusting for false discovery rate (q<0.05). *Ccl11* and *Duox2* are involved in osteogenic stem cell migration (*Smith et al., 2012*; *Tyurin-Kuzmin et al., 2016*), *Tnfaip6* and *Ccl7* in immunomodulation (*Li et al., 2022b*; *Rafei et al., 2008*), *Ptx3* is involved in both processes (*Cappuzzello et al., 2016*), and *Actg1* in cytoskeletal structure (*Dugina et al., 2009*; *Simiczyjew et al., 2014*). *Zbtb16*, encoding transcription factor PLZF (*Zbtb16*), regulates osteogenesis by acting upstream of master osteoblast transcription factor *Runx2* (*Figure 2—figure supplement 3i*; *Agrawal Singh et al., 2019*; *Liu and Lee, 2013*).

In the suture mesenchyme, analysis of 297 significantly upregulated genes in Aged CD90[+] cells during the IF diet compared to Aged AL control (*Figure 4b*) showed enrichment in 'Mitotic cell cycle' and 'Protein phosphorylation'. Notably, the proliferation marker Mki67 is significantly upregulated, with a threefold increase in Aged periosteum CD90[+] cells during intermittent fasting compared to ad libitum diet.

Intermittent fasting also showed an upregulation of *Dbp*, *Tef*, and *Hlf*, an effect that was conserved between osteoprogenitor compartments in Aged mice. (*Figure 2—figure supplement 3g–h*,

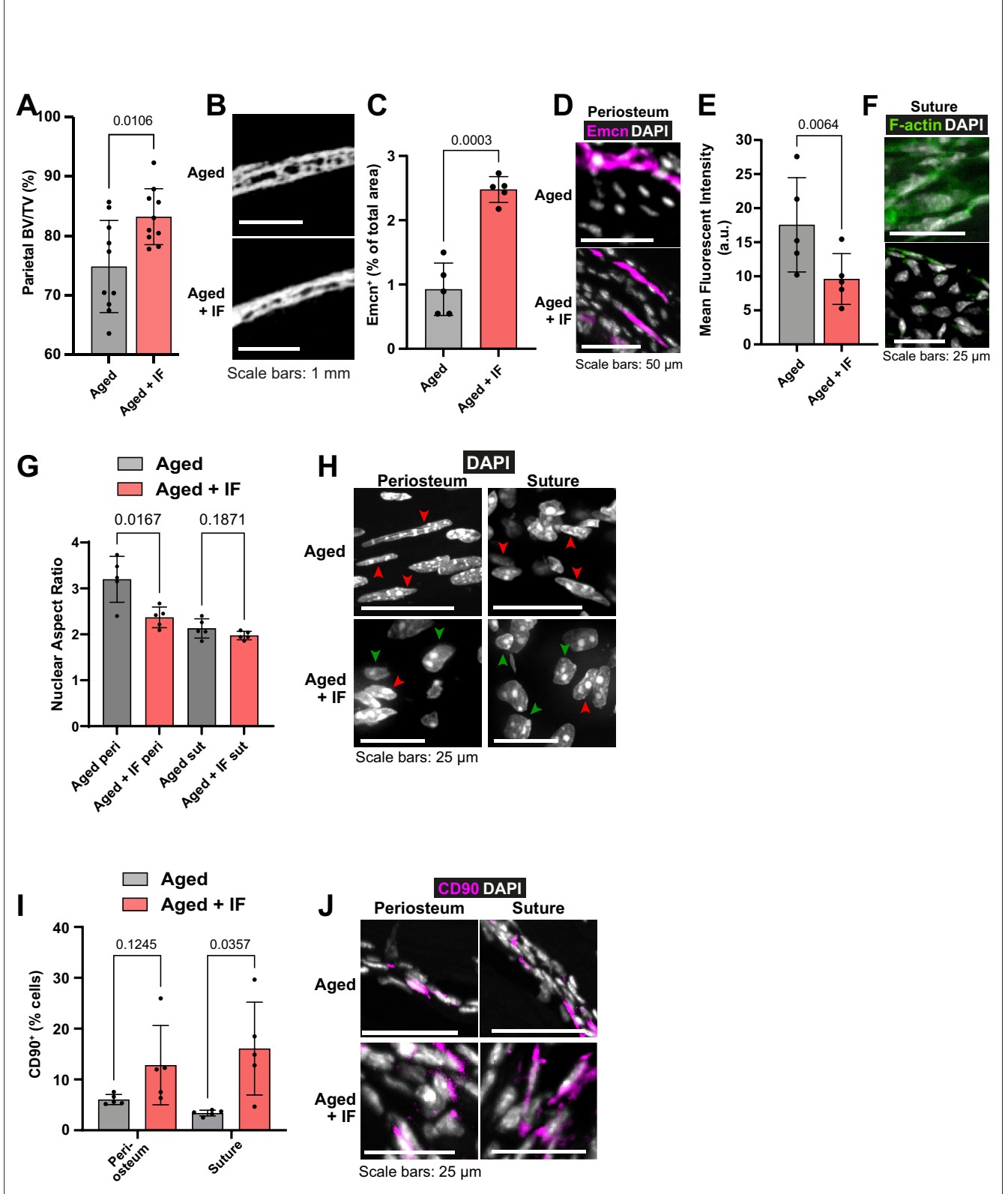

**Figure 3.** The impact of intermittent fasting on aged bone structure and osteogenic tissues. (**A**) Parietal Bone Volume/Total Volume (BV/TV) and (**B**) representative transverse micro-computed tomography (microCT) of Aged mice undergoing intermittent fasting. Statistical analysis: Welch's test in n=10 mice per condition. (**C**) Mean cross-sectional area taken up by Endomucin (Emcn)-positive vascular cells and (**D**) representative immunofluorescence staining of Emcn (magenta) and DAPI (grey) in the periosteum of Aged mice undergoing intermittent fasting. Statistical

*Figure 3 continued on next page*

*Figure 3 continued*

analysis: Welch's tests in n=5 mice per condition. (**E**) Mean Phalloidin-488 fluorescence intensity and (**F**) representative images for filamentous actin (F-actin, green) and DAPI (grey) in the suture mesenchyme of Aged mice undergoing intermittent fasting. Statistical analysis: Welch's test in n=5 mice per condition. (**G**) Nuclear aspect ratio (long axis divided by short axis) of cells within the periosteum and suture mesenchyme and representative DAPI-stained nuclei (**H**) of Aged mice undergoing intermittent fasting. Statistical analysis: Welch's tests in n=5 mice per condition. Green and red arrows highlight rounded and elongated nuclei, respectively. (**I**) Percentage of CD90+ cells within the periosteum (left columns) and suture (right columns), expressed relative to the total cellular number (DAPI+) in Aged mice undergoing intermittent fasting. Representative images (**J**) show CD90 immunostaining (magenta) and DAPI (grey). Statistical analysis: Welch's tests in n=5 mice. In panels A, C, E, G and I, columns in grey are data previously shown in *Figure 1*, reproduced here for reference. In panels B, D, F, H, J, Aged control representative images previously shown in *Figure 1* are reproduced here for reference.

The online version of this article includes the following figure supplement(s) for figure 3:

**Figure supplement 1.** Caloric Intake, Blood Glucose Levels, and Glucose Tolerance in Mice on Ad Libitum or Intermittent Fasting Diets.

*Supplementary file 2*). These genes are part of the transcriptional effectors of the circadian transcriptional network (*Gachon et al., 2004*), which act by binding to D-box sites in gene promoters (*Yoshitane et al., 2019*). In suture Aged CD90+ cells during the IF diet, *Dbp* alone is significantly upregulated against Aged AL CD90+ cells. Several other genes under circadian control (e.g. *Sirt1*, *Kat2b*, *Csnk1e*, *Ezh2*, *Fbxw11*, *Ucp2*; all p<0.05) are upregulated in Aged suture CD90+ cells during the IF diet. This corresponds to an enrichment of the transcriptional targets of *Clock/Per2/Arntl*, which together form the network of transcription factors responsible for maintaining circadian rhythms.

In *Figure 2a*, a shift in the periosteal CD90+ was noted with aging. Aged AL had relatively fewer osteogenic *Igf1*high cells, and more of *Igf1*low cells. Contrary, Aged +IF CD90+ cells were split 70.3% *Igf1*high, 29.7% *Igf1*low (*Figure 4c–d*), positioning them between Young and Adult distributions. 2 of the 14 genes identified as conserved markers of periosteal CD90+ cell aging (*Figure 2—figure supplement 4b*), *Vcan*, *Mmp14*, were significantly downregulated in Aged +IF. The proportion of 'proliferating' CD90+ suture cells increased in Aged +IF (*Figure 4e–f*). Further, 1 of the 32 conserved genes of the aging suture (*Figure 2—figure supplement 4c*), actin-associated *Tpm1* (*Li et al., 2022a*), was significantly downregulated in Aged +IF. *Wnt2*, found to be downregulated with aging, is significantly upregulated in the suture CD90+ population of Aged +IF (*Source data 2, supplementary table 10*).

To validate diet-induced changes in CD90+ cells, we conducted immunostaining for protein markers in histological sections. ATPB, marker of ATP synthase, appeared elevated in periosteum and significantly increased in suture mesenchyme CD90+ cells following IF (*Figure 4g–m*). This suggests an increase in cellular ATP levels, which is supported by decreases in ATP-deficiency sensor phospho-AMPK in periosteal and suture CD90+ cells (*Figure 4h and m*). While proliferating cells were absent in Aged mice, we observed Ki67+ cells in samples of the periosteum of 3 out of 5 Aged +IF mice and in 4 out of 5 Aged +IF mice for suture mesenchyme (*Figure 4i and m*). Conversely, p16 was similar in periosteum and suture mesenchyme (*Figure 4j and m*). Wnt/βcatenin pathway activity, as measured by the relative nuclear localization of β-catenin, was significantly increased within the CD90+ cells of both the periosteum and suture (*Figure 4k and m*). Lastly, expression of the motility marker MMP2 in CD90+ cells was markedly increased over Aged controls in the periosteum, and just under statistical significance in the suture mesenchyme (*Figure 4l and m*).

Together, these findings suggest that IF rejuvenates aged CD90+ cells, in part, by enhancing proliferation, immune response, ECM remodeling, Wnt/β-catenin pathway, and metabolism, including increased ATP levels and decreased AMPK levels.

## Short-term metabolic approaches rejuvenate the calvarial osteogenic compartments of Aged mice and promote bone repair

IF diet likely enhances energy supply to CD90+ cells and their function through various mechanisms. We next aimed to determine if specific underlying mechanisms alone could replicate the effects of IF on aged bone repair. Our analysis revealed that both aging and IF affect the NAD+ biosynthesis pathway and gut microbiota. Thus, we targeted these aspects to investigate their potential to enhance aged bone repair.

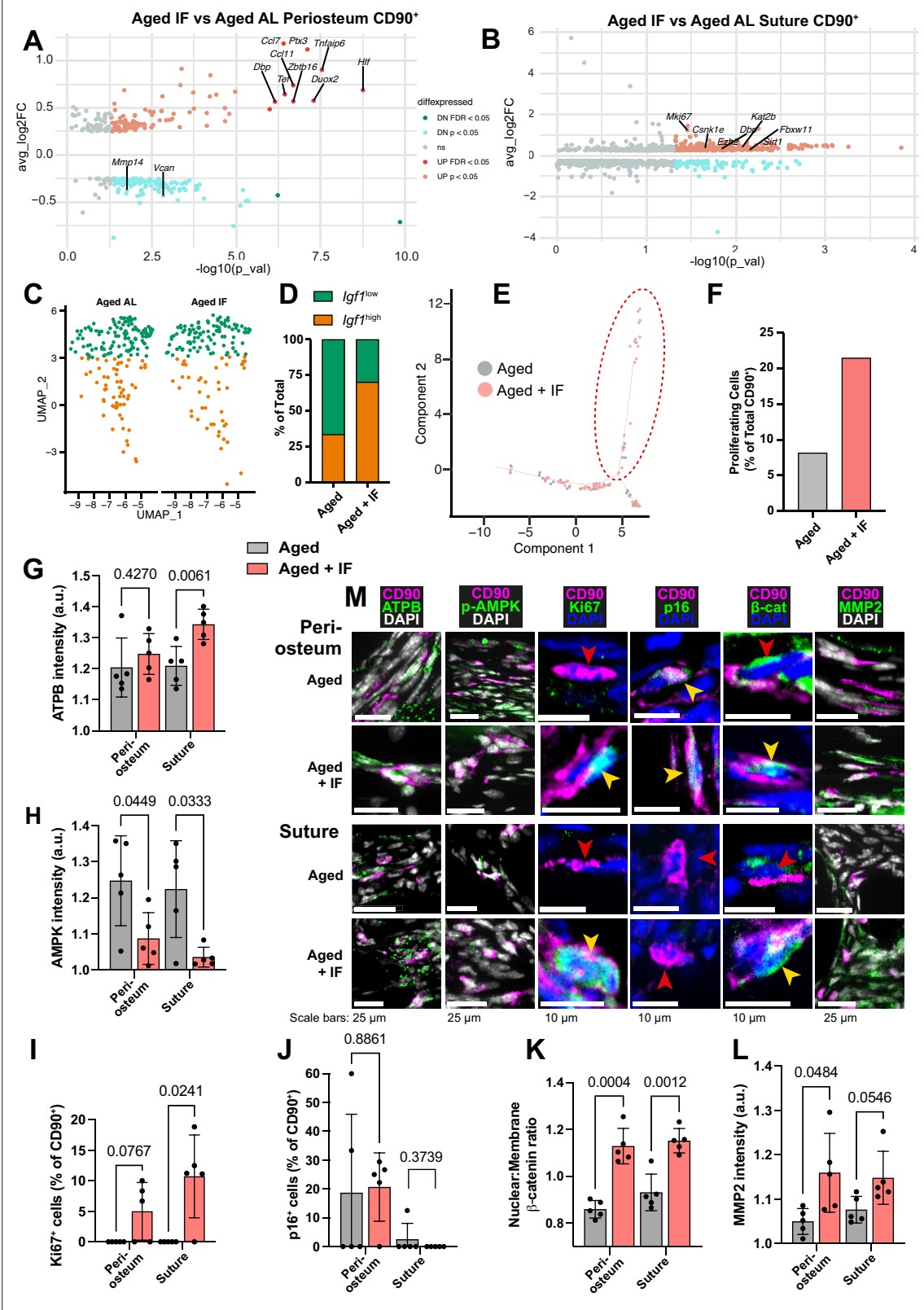

**Figure 4.** Transcriptomic and protein expression characterization of Aged CD90+ cells during intermittent fasting. (**A**) Volcano plot showing differentially-expressed genes in the comparison of the Aged CD90+ population in the periosteum and (**B**) suture during intermittent fasting (IF) vs the ad libitum (AL) diet. Statistical analysis: Wilcoxon Rank Sum test via the FindMarkers function in the Seurat R package. Upregulated (pink: p<0.05; red: FDR-adjusted p<0.05), downregulated (cyan: p<0.05; teal: FDR-adjusted p<0.05), and nonsignificant (ns) genes are displayed on axes of log fold expression

*Figure 4 continued on next page*

*Figure 4 continued*

(IF over AL) (y, avg_log2FC) against p transformed as its negative log value. Periosteum: Aged IF (n=194 cells); Aged AL (n=242 cells). Suture: Aged IF (n=79 cells); Aged AL (n=61 cells). (**C**) UMAP-plot showing the distribution of a subpopulation of CD90⁺ periosteal cells during intermittent fasting in Adult and Aged mice. AL cells are data previously shown, reproduced here for reference. (**D**) Percentage of the total cells categorized as either *Igf1*^high or *Igf1*^low. Aged +IF: *Igf1*^high (n=104), *Igf1*^low (n=44). (**E**) Plot (Monocle package) of suture CD90⁺ populations. (**F**) Percentage of CD90⁺ categorized as 'proliferating' (inside red dashed ellipse) vs 'non-proliferating' (outside ellipse) in Aged mice undergoing intermittent fasting. AL cells and chart are data previously shown, reproduced here for reference. Statistical analysis: two-tailed binomial test based on expected observed proportions between groups. Adult +IF: proliferating (n=30 cells), non-proliferating (n=86); Aged +IF: proliferating (n=17), non-proliferating (n=62). (**G**) Normalized intensity staining for ATPB in CD90⁺ cells. (**H**) Normalized intensity staining for phospho-AMPK in CD90⁺ cells. (**I**) Proportion of CD90⁺ cells that are Ki67⁺. (**J**) Proportion of CD90⁺ cells that are p16⁺. (**K**) Ratio of Nuclear:Membrane β-catenin staining in CD90⁺ cells. (**L**) Normalized intensity staining for MMP2 in CD90⁺ cells. For all plots (**G–L**), Aged +IF (red), Aged control (grey), periosteum (2 left columns), suture mesenchyme (2 right columns), statistical testing: Welch's tests in n=5 mice per condition. (**M**) Representative images depicting CD90 (magenta), DAPI (grey/blue as indicated by column), and functional markers (green) as indicated by column. Yellow arrows indicate marker-high/positive cells; red arrows indicate marker-low/negative cells. In panels C-M, data and representative images of Aged controls from previous figures are reproduced here for reference.

## NMN supplementation improves bone repair in Aged mice

NAD⁺ is a multifunctional enzyme co-factor with roles including protein deacetylation, histone modification, intracellular signaling, and redox reactions (*Covarrubias et al., 2021*). It is also crucial in oxidative phosphorylation (*Stein and Imai, 2012*), cycling between oxidized and reduced forms (NAD⁺/NADH) in the mitochondria as part of the electron transport chain. Our analysis of inferred NAD⁺-dependent pathway activity (*Figure 2—figure supplement 3*, *Source data 2, supplementary table 4*), live mitochondrial activity (*Figure 1i–j*), and markers of cellular energy levels (*Figure 2f, g and I*) demonstrated distinct changes to metabolism with age. Yet, little is known about the effect of NAD⁺-coupled cellular metabolism on the composition of aging calvarial periosteum and suture mesenchyme and its influence over Aged bone repair.

To increase cellular levels of NAD⁺ systemically, we orally dosed Aged mice with nicotinamide mononucleotide (NMN), an immediate precursor to NAD⁺ in the 'salvage' pathway (*Yang and Sauve, 2016*). During a pilot study involving oral NMN supplementation (600 mg/kg, three times within 1 week) in Adult mice, the mitochondrial membrane potential, as measured by TMRM staining, within the cells of the periosteum was significantly increased (*Figure 5a and b*). NMN was then administered to Aged mice for the 2 weeks prior and 2 weeks following calvarial defect surgery.

Parietal bone BV/TV was significantly increased in Aged animals that received NMN (Aged +NMN; *Figure 5c–d*). Remarkably, Aged bone repair was significantly increased to 54.32% with NMN with inactive bandages (*Figure 1o–p*), marking a threefold improvement over control Aged mice. Adding Wnt3a-bandage increased Aged +NMN repair to 71.18%, resulting in over 2.71 times greater area of new bone compared to Aged controls + Wnt3a-bandage, akin to repair in Adult mice +Wnt3a-bandage.

Next, we probed how NMN affects CD90⁺ cells in the osteogenic compartments. In the suture mesenchyme, NMN treatment significantly increased CD90⁺ cells compared to Aged control (*Figure 5e–f*). In the periosteum, CD90⁺ populations were unchanged. Additionally, compared to Aged controls, NMN supplementation significantly increased the vascularity of the periosteum (*Figure 5g–h*), and approximately halved actin cytoskeletal staining intensity in the suture mesenchyme (*Figure 5i–j*). Nuclei in the suture and periosteum were rounder and less elongated than Aged controls (*Figure 5k–m*), making them morphologically resemble Adult nuclei.

Taken together, NMN treatment increased periosteal vascularity, impacted aspects of tissue mechanics of the periosteum and suture mesenchyme, leading to a significant improvement in bone repair in Aged mice.

## Cyclical nutrient deprivation and NMN enhance mitochondrial function in osteogenic cells in vitro

Next, we aimed to model aspects of the effects of intermittent fasting and NMN supplementation on osteogenic cells, employing a cyclical nutrient deprivation regimen or NMN supplementation in vitro to elucidate aspects of their mechanisms of action. While this approach allows us to study specific cellular responses, it is crucial to recognize that it cannot fully replicate the complex systemic effects of in vivo intermittent fasting.

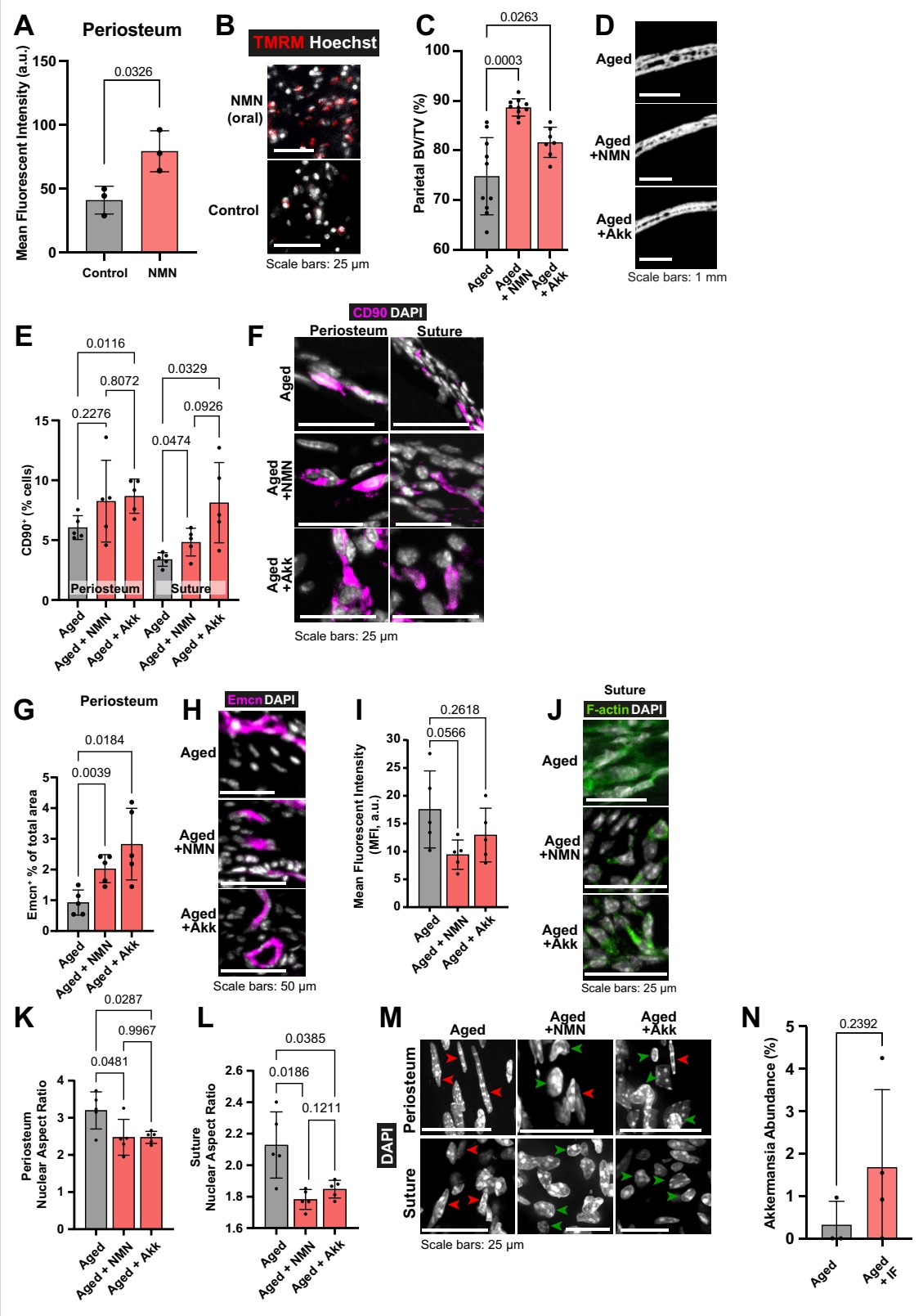

**Figure 5.** The impact of Nicotinamide mononucleotide (NMN) and *Akkermansia muciniphila* (Akk) supplementation on Aged bone structure and osteogenic tissues. (**A**) Mean intensity and (**B**) representative images of live TMRM staining (red) and Hoechst (grey) in Adult mice controls and Adult mice dosed with 600 mg/kg NMN by oral gavage, samples of periosteum ex vivo. Statistical analysis: Welch's test in tissue from n=3 mice per condition. (**C**) Parietal Bone Volume/Total Volume (BV/TV) and (**D**) representative transverse micro-computed tomography (microCT) sections in Aged mice

*Figure 5 continued on next page*

*Figure 5 continued*

supplemented with NMN or *Akkermansia*. Statistical analysis: Welch's tests in n=10 (Aged, Aged +NMN) and n=7 (Aged +Akk) mice. (**E**) Percentage of CD90$^+$ cells within the periosteum (left columns) and suture (right columns), expressed relative to the total cellular number (DAPI$^+$) in Aged mice supplemented with NMN or *Akkermansia*. Representative images (**F**) show CD90 immunostaining (magenta) and DAPI (grey). Statistical analysis: Welch's tests in n=5 mice per condition. (**G**) Mean cross-sectional area taken up by Endomucin (Emcn)-positive vascular cells and (**H**) representative immunofluorescence staining of Emcn (magenta) and DAPI (grey) in the periosteum of Aged mice supplemented with NMN or *Akkermansia*. Statistical analysis: Welch's tests in n=5 mice per condition. (**I**) Mean Phalloidin-488 fluorescence intensity and (**J**) representative images for filamentous actin (F-actin, green) and DAPI (grey) in the suture mesenchyme of Aged mice supplemented with NMN or *Akkermansia*. Statistical analysis: Welch's tests in n=5 mice per condition. (**K–M**) nuclear aspect ratio (long axis divided by short axis) of cells within the (**K**) periosteum and (**L**) suture in Aged mice supplemented with NMN or *Akkermansia*, and (**M**) Representative DAPI-stained nuclei. Statistical analysis: Welch's tests in n=5 mice per condition. Green and red arrows highlight rounded and elongated nuclei, respectively. (**N**) The relative abundance of bacteria identified as genus *Akkermansia* in 16S metagenomic profiling of Aged mice undergoing intermittent fasting vs. Aged controls. Statistical analysis: Welch's test in (Aged, n=3; Aged +IF, n=4) mice per condition. In panels C-M, data and representative images of Aged controls from previous figures are reproduced here for reference.

The online version of this article includes the following figure supplement(s) for figure 5:

**Figure supplement 1.** The relative abundance of Firmicutes to Bacterioides bacteria (**A**) and Lachnospiraceae bacteria (**B**), as measured by 16S metagenomic profiling.

We investigated the impact of these interventions on mitochondrial function in the MC3T3 calvarial osteoblastic cell line, a model commonly utilized to study osteogenesis in vitro and known for its ability to contribute to bone formation in vivo (*Horowitz et al., 1994*; *Paine et al., 2018*; *Wang et al., 1999*).

Alternating serum deprivation every hour (cyclical nutrient deprivation) in pyruvate and glucose-free medium induced the expression of genes related to mitochondrial biogenesis, fusion, mitophagy, and the unfolded protein response (*Figure 6a*). This was accompanied by increased mitochondrial filament length and membrane potential, indicating enhanced mitochondrial function (*McCarron et al., 2013*; *Westermann, 2012*; *Figure 6b–e*). These findings are consistent with our in vivo observations of improved mitochondrial function in animals subjected to intermittent fasting. While cyclical nutrient deprivation enhanced mitochondrial membrane potential, it also reduced overall cellular ATP levels. However, inhibiting mitochondrial ATP production with oligomycin (a Complex V inhibitor; *Devenish et al., 2000*) further diminished ATP levels, suggesting that mitochondrial respiration is a primary source of ATP in these cells (*Figure 6f*).

Recognizing the role of mitochondrial dysfunction in aging, we investigated the impact of NMN on MC3T3 cells challenged with oligomycin and cultured in glucose and pyruvate-free medium. NMN treatment mitigated oligomycin-induced ATP loss and ROS production, enhanced cell proliferation, and prevented the loss of mitochondrial membrane potential (*Figure 7a-c*). Holotomography microscopy revealed that NMN preserved the 'rod-like/filament' morphology of mitochondria in oligomycin-treated cells (*Figure 7d-e*). Furthermore, NMN restored the expression of genes related to mitochondrial biogenesis and dynamics, the unfolded protein response, and mitophagy, which were adversely affected by oligomycin (*Figure 7f*). Ultimately, NMN enhanced the growth and differentiation potential of oligomycin-treated cells, as evidenced by increased ALP activity and mineralization (*Figure 7g*).

In summary, our findings suggest that both cyclical nutrient deprivation and NMN supplementation can enhance mitochondrial function in osteogenic cells. NMN appears to protect against mitochondrial dysfunction induced by Complex V inhibition, highlighting its potential in mitigating the decline in osteogenic cell function.

## Akkermansia muciniphila supplementation improves bone repair in Aged mice

The 'gut-bone axis' links food intake, gut microbiome and bone metabolism via circulating metabolites (*Villa et al., 2017*). Bacteria in the genus *Akkermansia* appeared to be elevated during Aged +IF compared to the AL control diet (*Figure 5n*), consistent with previous studies that found similar results in young mice and human (*Li et al., 2020*; *Özkul et al., 2019*).

*Akkermansia muciniphila* is an anaerobic gut bacterium responsible for mucin turnover (*Derrien et al., 2008*), and a regulator of the intestinal mucus environment (*Reunanen et al., 2015*). The *Akkermansia* population exerts wider effects on the gut microbiome, and has been studied for its roles in

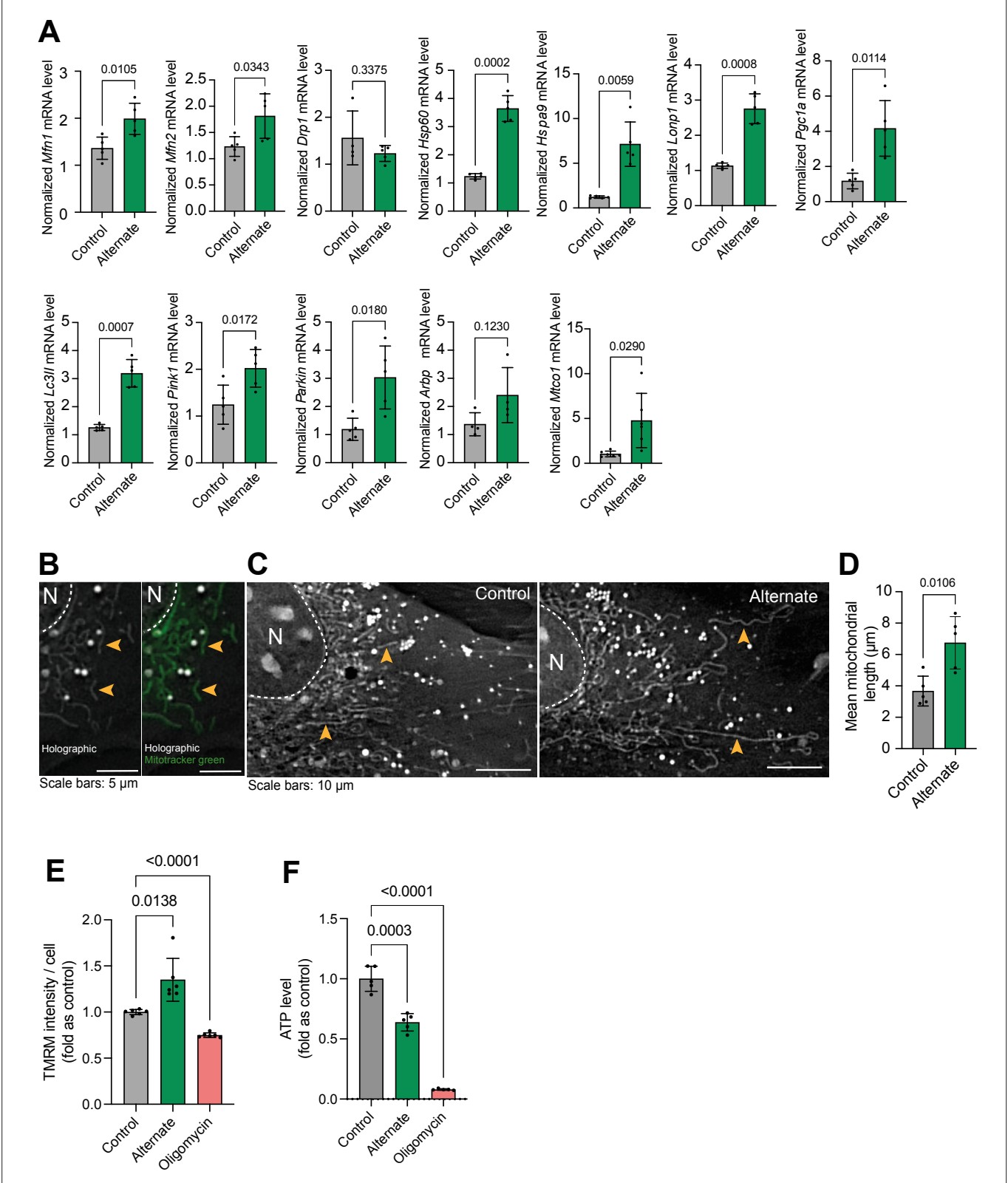

**Figure 6.** Effect of hour-based alternate serum deprivation on mitochondria in MC3T3 cells in glucose-free media. (**A**) Effect of alternate serum deprivation on gene expression involved in mitochondrial dynamics (*Mfn1*, *Mfn2*, *Drp1*), unfolded protein response (*mtUPR*) (*Hsp60*, *HspA9*, *LonP1*), biogenesis (*Pgc1a*), structure (*Arbp*), mitophagy (*Lc3II*, *Pink1*, *Parkin*), and mitochondrial respiratory complex IV (*Mtco1*). Statistical analysis: Welch's test, n≥4/group. (**B**) Representative holographic images of mitochondrial shape (orange arrowheads), confirmed by Mitotracker green staining. (**C–D**) Effect

*Figure 6 continued on next page*

*Figure 6 continued*

of alternate serum deprivation on mitochondrial length. Statistical analysis: Welch's t-test of n=5/group, total mitochondria = 425. (**E**) Effect of alternate serum deprivation and oligomycin on mitochondrial membrane potential. Statistical analysis: Welch's t-test of n=6/group with at least 481 cells/group. (**F**) Effect of alternate serum deprivation and oligomycin on ATP level. Statistical analysis: Welch's t-test of n≥5/group.

metabolic health (*Dao et al., 2016*; *Leite et al., 2021*). We cultured *Akkermansia muciniphila* (Akk) under anaerobic conditions and calculated the viability of cryopreserved aliquots. Doses of live Akk (2x10$^8$ C.F.U.) were given by oral gavage three times per week in Aged mice for the 2 weeks prior and 6 weeks following calvarial defect surgery.

Our metagenomic analysis reveals that Firmicutes:Bacteroides ratio, indicator of microbiome composition (*Magne et al., 2020*), is significantly increased in Aged mice with *Akkermansia* supplementation (Aged + Akk; *Figure 5—figure supplement 1a*). Furthermore, the proportion of gut bacteria belonging to family *Lachnospiraceae,* indicative of the capacity of the gut bacteria to produce short chain fatty acids (SCFAs), is significantly elevated in Aged + Akk (*Figure 5—figure supplement 1b*). SCAFs enter the host bloodstream and exert epigenetic effects primarily through histone modification, which might have implication on cell function and differentiation (*Licciardi et al., 2011*).

Similar to NMN treatment, parietal bone BV/TV was significantly increased Aged +Akk (*Figure 5c–d*). Aged bone repair was significantly increased to 43.76% with Akk and inactive bandages (*Figure 1o–p*). In combination with the Wnt3a bandage effect, bone repair in Aged +Akk further increased to 64.84%. This level of bone repair is similar to that of Young mice with Wnt3a-bandage (*Supplementary file 1*). Aged +Akk also produced striated bone, similar in structure to Young mice, under both Wnt3a and control bandages (*Figure 1—figure supplement 3d*).

Aged +Akk led to a significant increase in CD90$^+$ cells in the periosteum and suture mesenchyme (*Figure 5e–f*) and a threefold increase in the Emcn$^+$ vascularity area of the periosteum (*Figure 5g–h*). Compared to Aged controls, in Aged +Akk, actin cytoskeletal staining intensity appeared reduced (not significant) in the suture mesenchyme (*Figure 5i–j*), and nuclei in both the suture and periosteum appeared rounder and less elongated (*Figure 5k–m*). Accordingly, the nuclear morphology of Aged +Akk cells resembles that of younger mice.

In summary, we have revealed a multifaceted decline in CD90$^+$ cells with aging, which can be locally counteracted by a Wnt3a-bandage or systemically enhanced through an IF diet. Moreover, systemic interventions, acting via the gut and on shorter terms, can improve CD90$^+$ populations, osteogenic compartments, and significantly enhance bone repair in Aged mice. Combining the Wnt3a-bandage with systemic interventions targeting cellular energy supply, proliferation, migration, and Wnt signaling presents coordinated and clinically relevant approaches, promising rejuvenation of Aged bone repair comparable to younger individuals.

## Discussion

Here, we report on systemic interventions that can reverse aspects of the aging osteogenic tissue to promote effective calvarial bone repair (*Figure 7—figure supplement 1*).

While extensive research focuses on long bones (*Yates et al., 2007*), there is a significant gap in understanding aging in flat bones. Our study on the calvaria demonstrates a clear decline in both bone structure and healing capacity with age. We observed a progressive decline in periosteal blood vessel density beginning in adulthood. Microvascular rarefaction is increasingly recognized as a driver, rather than merely a consequence, of hallmarks of cellular aging, such as cellular senescence (*Goligorsky, 2010*; *Lähteenvuo and Rosenzweig, 2012*). Indeed, previous studies inducing vasculature loss in young animals accelerates cellular aging (*Chen et al., 2021*), while mitigating this vascular decline preserves tissue health and delays aging (*Grunewald et al., 2021*). Vascularization and modulation of blood flow are critical for calvarial bone repair (*Holstein et al., 2011*; *Huang et al., 2015*; *Ren et al., 2024*). Early in the repair process, periosteal blood vessels grow independently of osteoprogenitor cells, establishing a supportive environment that promotes osteoprogenitor migration and subsequent ossification (*Bixel et al., 2024*). Compromised vascularization significantly impairs the healing of critical-sized calvarial defects (*Caliaperoumal et al., 2018*).

Aging bone tissues are inherently more hostile and pro-apoptotic (*Almeida, 2012*), leading to limited success with cell-based repair approaches in aged individuals (*Hirata et al., 2022*; *Liu*

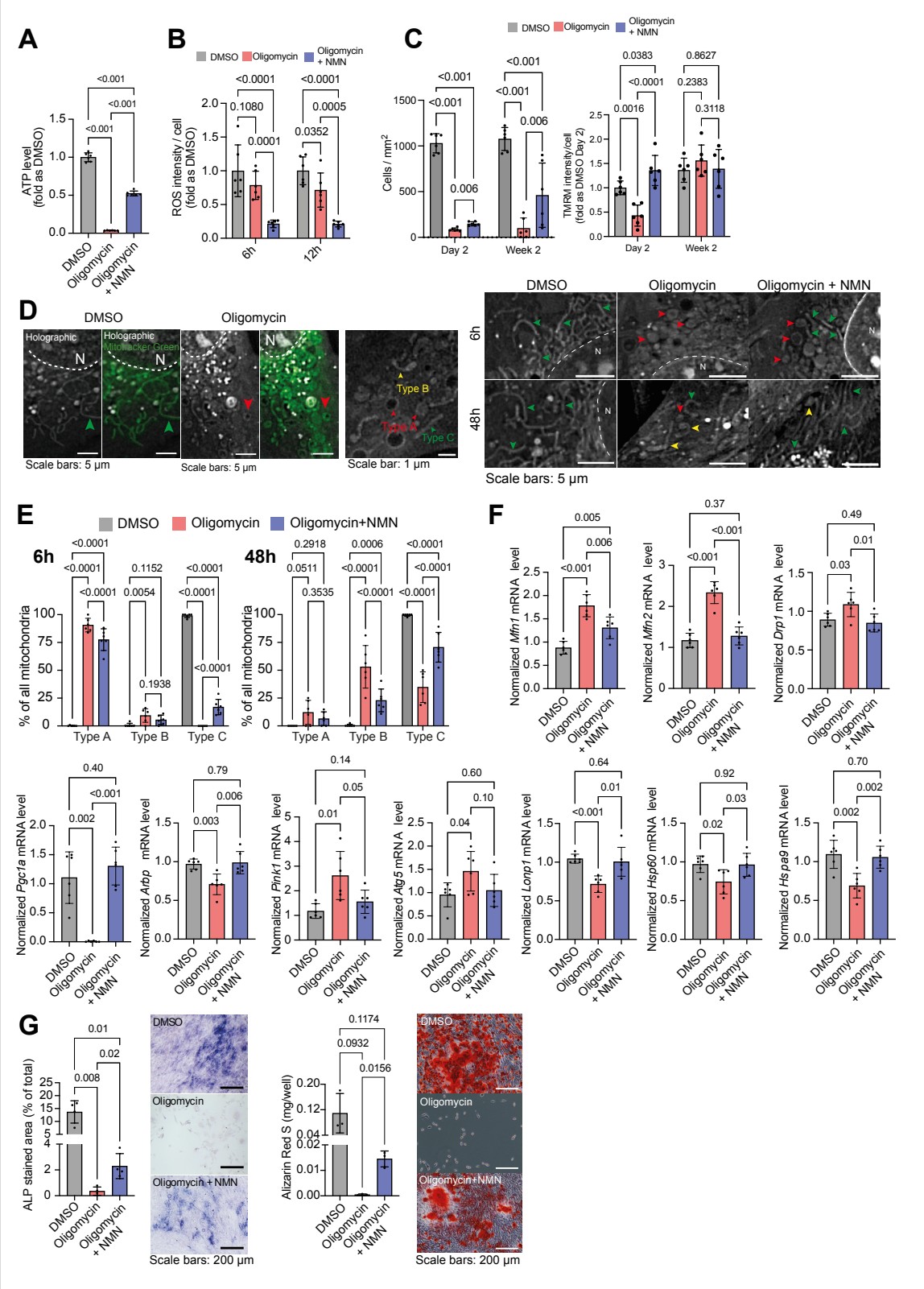

**Figure 7.** Effect of NMN on oligomycin-induced mitochondrial damage in glucose-free MC3T3 cells. (**A**) Effect of oligomycin ±NMN on ATP level in MC3T3 cells. Welch's test, n=6/group. (**B**) Effect of oligomycin ±NMN on ROS level MC3T3 cells. Welch's test, n=6/group. (**C**) Effect of oligomycin ±NMN on cell density (left) and mitochondrial membrane potential (right) over time in MC3T3. Welch's test, n=6/group. (**D**) Representative images of mitochondrial shape alteration by oligomycin exposure ±NMN supplementation. Dashed white line = nuclear membrane, N=nucleus. Observation

*Figure 7 continued on next page*

*Figure 7 continued*

by holotomography (grey levels) and confirmed with Mitotracker Green (green). Red arrowheads = Type A ('round'), yellow arrowheads = Type B ('intermediate'), and green arrowheads = Type C ('rod-like/filamentous') mitochondria. (**E**) Quantification of the effect of oligomycin ±NMN on mitochondrial shape. Each data point is a cell with at least 30 mitochondria. Welch's test n=6 cells / group, total mitochondria = 1821. (**F**) Effect of oligomycin ±NMN on gene expression involved in mitochondrial dynamics (Mfn1, Mfn2, Drp1), biogenesis (Pgc1a), structure (Arbp), mitophagy (Pink1, Atg5), and unfolded protein response (mtUPR) (LonP1, Hsp60, HspA9). Welch's test, n≥5/group. (**G**) Effect of oligomycin ±NMN on osteoblastic differentiation. Staining of alkaline phosphatase (ALP) activity and mineralization (Alizarin Red S; ARS) of the matrix after two weeks of differentiation. Welch's test, n=4 (ALP) or 3 (ARS) / group. *Figure 1—figure supplement 1*. (**A**) Methodology for segmentation of micro-computed tomography (microCT) images for calculation of bone morphometry metrics: (**B**) Parietal Thickness (**C**) Cortical Tissue Mineral Density (Ct.TMD), (**D**) Parietal Whole Bone Mineral Density (BMD). Statistical analysis: Welch's tests, n = 10 mice per condition, except Aged + Akk (n = 9). (**E**) The method for quantifying immunofluorescence-positive (red) cells. The maximum pixel intensity is determined in control tissues lacking primary antibody (no primary control). This value is used to threshold the stained sample tissue, with the remaining signal distinguishing positive cells (red arrows). DAPI (grey) is also shown for reference of cellular locations. (**F**) Number and percentage of blood vessels positive for Emcn and/or CD31 with representative images in the periosteum and suture mesenchyme. (**G**) Methodology for segmentation of Endomucin (Emcn) immunofluorescent staining in fixed tissue sections, for calculation of vascular histomorphometry metrics: (**H**) Emcn$^+$ staining density (proportion of total tissue area). Data are shown previously in main *Figures 1, 3 and 5*, and are reproduced here for reference, (**I**) Mean lumen area per blood vessel, (**J**) Number of blood vessels per area tissue (BV.N). Statistical analysis: Welch's tests, n = 5 mice per condition. (**K**) Representative brightfield images of unstained tissue sections highlighting the overlying periosteum (red P, red dashed line) and parietal bone (red B). (**L**) Mean periosteal thickness. Statistical analysis: Welch's tests, n = 4 mice per condition, except Aged + Akk (n = 3).

The online version of this article includes the following figure supplement(s) for figure 7:

**Figure supplement 1.** Schematic that recapitulates the mains findings of the study.

---

et al., 2018). Stiffness increases with aging in various tissues (*Ferrari and Pesce, 2021*; *Lim and Ito, 2022*; *Segel et al., 2019*), impacting cytoskeletal configuration (*Khan et al., 2020*; *Pongkitwitoon et al., 2016*) and significantly affecting osteogenic differentiation and senescence in vitro (*Liu et al., 2022*). Our study shows increased actin expression and elongated nuclear shape in aging osteogenic compartments, alongside elevated levels of *Mmp9* and *Mmp14*, linked to stiffer osteogenic tissues (*Tang et al., 2013*; *Zhu et al., 2020*). These changes are associated with a decline in both the number and function of osteoprogenitors, including impaired proliferation, mitochondrial energy metabolism, Wnt/β-catenin pathway activation, migration, and ultimately, bone defect repair. While Wnt-bandage treatment increases the number of osteoprogenitors in the defect site across all age cohorts, it only modestly enhances bone repair in aged animals compared to the healing observed in adult mice.

We show that interventions improving whole-body energy metabolism can enhance aged bone health and repair. While IF is effective in delaying aging in adult flies, animals need to switch back to ad libitum feeding at older ages (*Catterson et al., 2018*), and increased protein intake is suggested for optimizing health span and longevity in aged mice and humans (*Levine et al., 2014*). Our findings, however, reveal a distinct paradigm for IF in aged bone repair. Specifically, short-term IF initiated post-injury robustly enhances bone healing in aged animals, comparable to younger counterparts.

Mechanistically, IF exerts its effects through several pathways, including enhanced stress resistance, increased reliance on lipid fuel utilization, and activation of proteostatic mechanisms, all contributing to delayed aging (*Longo and Anderson, 2022*). Although we did not observe distinct changes in overall signature of lipid metabolism or glycolysis in osteogenic tissues during aging or IF, we did find a decline in mitochondrial function and ATP supply with age. Our in vitro studies showed that alternating cycles of serum deprivation and supply in osteogenic cells induced genes related to the unfolded mitochondrial protein response, mitophagy and mitochondrial fission associated with increased mitochondrial membrane potential and mitochondrial fission, indicating improved function (*McCarron et al., 2013*; *Westermann, 2012*).

Our transcriptomic analysis shows that *Mmp14* is reduced during IF in the periosteum. In the suture mesenchyme, *Piezo1*, a gene known to mediate mechanically induced aging (*Ren et al., 2022*; *Segel et al., 2019*), was also significantly reduced in aged mice undergoing IF. Along with the reversal of nuclear cell morphology and levels of F-actin, these results indicate that IF also impacts the mechanics of the tissue, making it resemble that of younger animals. IF also upregulated *Dbp*, *Tef*, and *Hlf*, genes involved in the circadian transcriptional network (*Gachon et al., 2004*), in both osteoprogenitor compartments of aged mice. These genes, which act on D-box sites, are linked to premature aging when knocked out, affecting the liver, kidney (*Gachon et al., 2006*), and hematopoietic stem cell differentiation. While circadian rhythms' impact on bone is mostly studied in osteoblasts in vitro

(*Luo et al., 2021*; *Zhou et al., 2018*), our data suggest that circadian transcription may play a role in rejuvenating CD90[+] cells for bone formation and repair in aged mice.

ChIP-SEQ studies *Yoshitane et al., 2019* have identified D-box sites near promoters of Wnt/β-catenin components (e.g. *Lrp5/6*, *Wnt8a*, *Fzd4*) and osteogenic transcription factors like *Zbtb16*, linking circadian rhythms, stem cell function, and Wnt signaling (*Lecarpentier et al., 2014*; *Matsuura et al., 2018*). While food intake regulates circadian rhythms in several organs (*Damiola et al., 2000*; *Pickel and Sung, 2020*), the diet-circadian interaction in bone repair is not well understood. Investigating circadian transcription's role in enhancing osteoprogenitor function for bone repair in aged mice is promising.

IF also led to beneficial shifts in gut microbiota composition, including elevated levels of *Akkermansia muciniphila* and increased expression of genes involved in NAD[+] biosynthesis. Subsequently, we show that short-term supplementation with *Akkermansia muciniphila* or NMN in aged mice significantly improve periosteal blood vessel density, osteoprogenitor number and function, osteoclast activity and enhance bone health and repair.

Although IF, Akk or NMN can act through many pathways and cellular processes, previous studies highlighted their beneficial effect on metabolism, particularly their influence on NAD-dependent pathways. Mechanistically, our in vitro studies using calvarial osteogenic cells revealed that NMN protects against mitochondrial damage, often associated with aging, promotes healthy mitochondrial morphology and function, and enhances osteogenesis.

The effects of the three metabolic approaches on bone health and repair have not been studied in the context of already-aged individuals. NMN shows protective effects against bone health decline, but in long-term studies that begin dosing well before old age (*Kim et al., 2021*; *Mills et al., 2016*; *Song et al., 2019*). The impact of *Akkermansia muciniphila* on bone health or healing has only been investigated in young adult models with subcritical defects (*Lawenius et al., 2020*; *Liu et al., 2020*). IF diet has been investigated for skin wounding (*Luo et al., 2020*) in young animals but not for bone repair. We present key progress in developing three clinically relevant approaches for bone tissue regeneration in elderly patients. First, we describe the effects of NMN or *Akkermansia* dosing, or IF diet on bone tissues in aged animals that have already experienced severe skeletal decline. Second, our rejuvenation protocols are applied for shorter terms, and even post-injury with IF diet.

This study offers valuable insights into bone aging and rejuvenation strategies. However, future research should address several limitations. Further investigation is needed on the mechanisms of action of IF, NMN, and *Akkermansia*, optimal intervention timing and duration, and synergistic effects of combined approaches. While this study focused on a common osteoprogenitor, investigating the roles of specific skeletal stem cell populations and other cells within the osteogenic compartments (including immune cells) in flat bones represents a compelling frontier.

Our findings suggest broader implications beyond bone health, with these rejuvenation strategies potentially translating to other aging organs and injuries. Rigorous investigation of dosage, safety, and long-term effects will be essential for human applications.

# Materials and methods

## Wnt3a bandage manufacture

Polycaprolactone (PCL)-based polymer film (Poly-Med) was plasma-treated using a plasma etching electrode in oxygen gas at a flow of 20 cm$^3$ min$^{-1}$ with a radio frequency plasma generator (frequency 13.56 MHz, power 50 W, Diener electronic Zepto-W6) at 0.2 mbar pressure for 3 min. Treated films were immediately incubated in 5% (3-Aminopropyl)triethoxysilane (APTES, Sigma, 440140) in ethanol (dark, room temperature, 2 hr), before washing twice with 100% ethanol (room temperature, 10 min), and allowed to air dry. Circular bandages were cut using a 5 mm biopsy punch (Stiefel) and placed in 10 mg/ml succinic anhydride (Sigma Aldrich, 239690–50 G) in 0.4 M borate buffer (Sigma-Aldrich, B7660) pH = 9 (room temperature, 30 min). Bandages were twice washed in 25 mM Morpholineethanesulfonic acid (MES) buffer pH=5 (Sigma-Aldrich, M3671), then incubated in a MES buffer solution containing 50 mg/ml N(3Dimethylaminopropyl)-N'-ethylcarbodiimide hydrochloride (EDC, Sigma-Aldrich, E7750) and 50 mg/ml NHydroxysuccinimide (NHS, Sigma-Aldrich, 56480) (room temperature, 30 min), and washed twice again in MES buffer. Bandages are individually placed on a sterile flat

surface and briefly allowed to air-dry. 300 ng Wnt3a (R&D, 1324-WN-010) was diluted to 23 µL in MES buffer, and this liquid added as a droplet on top of the bandage for 1 hr at room temperature.

For inert control bandages (inactive-bandages), the bandages were then placed into a 96 well plate and washed with PBS. 20 mM Dithiothreitol (DTT, Roche, 10197777001) in 50 µL ddH$_2$O was then added and incubated (37°C, 30 min). Active Wnt3a-bandages were also placed in the 96-well plate, and then all bandages were washed with PBS, and quenched with 2% Fetal Bovine Serum (FBS, Sigma-Aldrich, F7524) in Dulbecco's Modified Eagle Medium (DMEM, Thermo Fisher, 41966; room temperature, 15 min) in preparation for their surgical implantation.

### Wnt3a-bandage testing

The Wnt/β-catenin pathway activation of Wnt3a-bandages was confirmed as previously published (*Lowndes et al., 2017*). Wnt3a-bandages were tested for Wnt/β-catenin pathway activation using Comma-D Beta cells harboring GFP expression under the control of a TCF-LEF promoter (*Lowndes et al., 2016*; *Figure 1—figure supplement 3a, c*). Similarly, LS/L cells harboring luciferase under control of a 7xTCF promoter were used to quantitatively test Wnt3a-bandages, as described previously (*Okuchi et al., 2021*; *Figure 1—figure supplement 3a, c*).

### Animal experimentation methods according to the ARRIVE Essential 10 guidelines

All procedures involving animals were carried out by holders of Personal Licences (PIL) under a Project Licence (PPL number: P8F273FDD, S.J.H.) shaped by ARRIVE and NC3Rs guidelines. The PPL was approved by the Animal Welfare and Ethical Review Body (AWERB) at King's College London and granted by the Home Office under the Animals (Scientific) Procedures Act 1986 (ASPA) and Amendment Regulations 2012.

### Study design

Within each animal, the bone repair under the Wnt-bandage was compared to the one under the inactive bandage wherever appliable. Between groups of treatments, the bone repair was compared to other groups, or to a control group (sham treatment/ad libitum feeding). Each animal was used as an experimental unit.

### Sample size

10 animals were used per surgical group for a total of 60 animals. The minimum is 7 animals/ group to get statistically reliable tests of the hypothesis (80% power with 0.05 type I error). We applied the resource equation method (*Chan et al., 2013*). Based on our experience 8–10 animals were used per group, to account for morbidity and animal variability (*Okuchi et al., 2021*).

### Inclusion and exclusion criteria

All animals were included in the study. Animals that were scarified/found dead before the end of the study were excluded (two for the group 'Adult' (n=8), and one for the groups 'Aged' (n=9) and 'Aged +Akk.' (n=9), n=10 in all other groups).

### Randomization

Randomization method was not applied to assign animals to experimental groups. Animals were identified using ear notches.

### Blinding

The experimenter was aware of the group allocation of the animals when performing the experiments and during the data acquisition and analysis.

### Outcome measures

Bone repair was assessed using microCT as described in the Micro-Computed Tomography section, and subsequent histological analysis and immunology staining were performed on cryosections as described in the Sample Processing and Cryosectionning, and Fluorescence Immunostaining sections.

## Experimental animals

Female mice were chosen for their delayed healing and higher fracture risk (*Haffner-Luntzer et al., 2021*; *Ortona et al., 2023*), leading to a more challenging case for bone repair, making them a stringent test for evaluating the effectiveness of our rejuvenation approaches. Female C57BL/6 J mice (Charles River, strain 632) were housed in conditions that comply with the Animals (Scientific) Procedures Act 1986 (ASPA). Mice were group housed (2–5 mice/cage) under 7 a.m.-7 p.m. lighting and fed with standard mouse food pellets (LabDiet, 5053). Three age groups were used for this study: Young (6 weeks), Adult (13 weeks), and Aged mice (>88 weeks).

For ad libitum diet, the food was renewed every 3–4 days. During intermittent fasting, food was removed from cages at 10 a.m. on Monday, Wednesday, and Friday, and returned at 10 a.m. on Tuesday, Thursday, and Saturday, such that the maximum time without food was 24 hr. Intermittent fasting started the day after the surgery and was performed for 10 weeks.

## Caloric intake calculation

To assess the caloric intake of mice, the food was weighted when made available to the mice ($W_{in}$), and when removed ($W_{out}$). The daily consumed food was calculated based on the weight difference ($W_{in} - W_{out}$), then converted to kcal (1 g=3.02 kcal, LabDiet, 5053), and expressed as kcal/mouse/day for each cage (n cage ≥3 with 1–5 mice/cage).

## Blood glucose level measurement

Fasting blood glucose levels were measured (Accu-Check tests strips) from 6 hr fasting mice by blood sampling the tail vein. For intraperitoneal glucose tolerance test (IPGTT), glucose was injected intraperitoneally (2 g/kg), and the blood glucose levels were measured after 15, 30, 60, and 120 min.

## Experimental procedures

### Calvarial defect

All steps of surgery were carried out under aseptic conditions. Mice were intraperitoneally anaesthetized with 10 µL/g bodyweight of anesthesia cocktail (7.5 mg/ml Ketamine hydrochloride (Ketavet) and 0.1 mg/ml Medetomidine hydrochloride (Domitor) in 0.9% sterile saline). Loss of consciousness was confirmed with lack of pedal and tail reflexes. The top of the head was shaved, cleaned, and prepared for surgery, before an ~8 mm central incision was made in the skin with surgical scissors. The periosteum was gently pushed towards the outer edges of the calvaria, and the location of the sutures noted. A 2 mm diameter full thickness defect was drilled into the middle of the parietal bone on both sides of the skull, ensuring irrigation was applied to limit heat buildup. A small amount (~10 µL) of 1 mg/ml collagen gel (Corning, 354249; diluted with DMEM and set with 3 mM sodium hydroxide (Sigma-Aldrich, S5881)) was placed into the defect to exclude trapped air, and the required bandages were placed face down over the defect. Small dots of glue (3 M, Vetbond) were applied at the edges of the bandages to affix them to the surface of the skull. The skin was then approximated and glued to ensure closure of the wound. Mice were administered 10 µL/g bodyweight of the recovery cocktail intraperitoneally (0.125 mg/ml Atipamezole hydrochloride (Antisedan) and 0.01 mg/ml Buprenorphine hydrochloride (Vetergesic) in 0.9% sterile saline) and placed in a warmed cage (35 °C). Mice typically became conscious within 20 min, began moving around the cage within 40 min, and were feeding within 2 hr. Mice showed no signs of distress or pain the following day and were externally monitored daily for the healing of the skin wound. Following surgery, mice were maintained under standard conditions for 10 weeks, and group housed (2–5 mice/cage) where possible.

### NMN supplementation

600 mg/kg bodyweight doses of β-nicotinamide mononucleotide (NMN, APExBIO, B7878), dissolved in water were given by flexible-tipped oral gavage 3 times per week, not on consecutive days, from 2 weeks before surgery to 2 weeks after, a total of 4 weeks.

### Anaerobic culture of *Akkermansia muciniphila*

Preserved *Akkermansia muciniphila* (Collection de l'Institut Pasteur, CIP107961T) was resuspended and cultured in Gifu's Anaerobic Medium (GAM) broth (Nissui Pharmaceutical, 05422) in an anaerobic environment (80% Nitrogen, 10% Carbon dioxide, 10% Hydrogen). Aliquots were cryopreserved in

50% glycerol (Sigma-Aldrich, G2025) in PBS. Post-thawing viability was determined by serial dilution of cryopreserved aliquots, plating onto GAM-Agar (Sigma-Aldrich, 01916) plates, and 24 hr anaerobic culture.

### *Akkermansia muciniphila* supplementation

Cryopreserved doses of $2.0 \times 10^8$ C.F.U. *Akkermansia muciniphila* were warmed to ~30°C and administered to Aged mice by flexible-tipped oral gavage 3 times per week, not on consecutive days, from 2 weeks before surgery to 6 weeks after, a total of 8 weeks.

## Results

Results are displayed as mean ± SD. All numerical values are available in the Source Data files.

## Ethical approval for animal experimentation

All procedures involving animals were carried out by holders of Personal Licences (PIL) under a Project Licence (PPL, P8F273FDD, S.J.H.) shaped by ARRIVE and NC3Rs guidelines. The PPL was approved by the Animal Welfare and Ethical Review Body (AWERB) at King's College London and granted by the Home Office under the Animals (Scientific) Procedures Act 1986 (ASPA) and Amendment Regulations 2012.

## Microbiome profiling

Feces samples were freshly collected and snap-frozen in liquid nitrogen. Samples were submitted for full length (V1-V9) 16 S metagenomic profiling. Amplification with KAPA HiFi HotStart ReadyMix PCR Kit (KAPA) and 16 S barcode primers (PacBio) preceded library preparation SMRTbell Express Template Prep 2.0 (PacBio). Samples were sequenced via the Illumina MiSeq to generate amplicon sequence data specifying the V3-V4 regions of the 16 S rRNA ribosomal subunit. The metagenomic analysis was conducted using the amplicon sequence variant (ASV) method of taxonomic classification facilitated by DADA2 (*Callahan et al., 2016*) incorporated by the Qiime2 package (*Bolyen et al., 2019*). Quality control metrics of the amplicon data was visualized by FastQC (*Andrews, 2015*). The data was cleaned, via removal of adapters and low-quality sequences, using Trimmomatic (*Bolger et al., 2014*) and Qiime2.

## Micro-computed tomography

Mice were culled by cervical dislocation and their tissues immediately collected. Whole calvarial samples were fixed in 4% PFA (room temperature, overnight) and washed in PBS. Fixed samples were scanned on a μCT50 micro-CT scanner (Scanco) in 19 mm scanning tubes, using voxels of side length 10 μm, X-ray settings of 70 kVp and 114 μA and using a 0.5 mm aluminum filter to attenuate harder X-rays. Scans were reconstructed and calibrated using known densities of hydroxyapatite (HA) between 0 and 790 mg HA $cm^{-3}$, with absorption values expressed in Hounsfield Units (HU). 3D volumes were constructed from raw data in SCANCO Visualizer 1.1, and signal of less than 250 mg HA $cm^{-3}$ was considered non-mineralized and was set as the black level in displayed images.

## Sample processing and cryosectionning

Fixed samples were decalcified in 10% formic acid (room temperature, overnight), then washed in PBS. Dehydration was carried out in 30% sucrose/PBS (room temperature, overnight), followed by 30% sucrose/30% OCT/PBS (room temperature, overnight). Samples were embedded in 100% OCT solidified with dry ice and stored at –80°C. Cryosections were cut at 14 μm thickness using a cryostat microtome at –23°C. Cut sections were stored at –20°C.

## Fluorescence immunostaining

Sections were rehydrated to ddH$_2$O. Antigen retrieval was performed with 0.05% v/v Tween-20 (Sigma-Aldrich, P9416) in 10 mM Tris (Sigma-Aldrich, 93362), 1 mM EDTA (Sigma-Aldrich, E6758), pH = 9.0 (40°C, 30 min). Sections were then permeabilized with 0.1% Triton X-100 (Sigma-Aldrich, 93443) in 1% Bovine Serum Albumin (BSA, Sigma-Aldrich, A2153) in phosphate buffered saline (PBS, Sigma-Aldrich, P4417) (1% BSA/PBS) (room temperature, 5 min), washed with 1% BSA/PBS, and individual staining regions were designated with hydrophobic delimiting pen (Agilent, S200230-2).

Primary antibodies were diluted in 0.05% Triton-X in 1% BSA/PBS and incubated in a humid chamber (4°C, overnight). Triplicate washes of 0.1% Triton-X in 1% BSA/PBS were carried out (room temperature, 10 min). Secondary antibodies were diluted in 0.05% Triton-X in 1% BSA/PBS, with 5 µg/ml DAPI (Sigma-Aldrich, D9542), and 1:400 dilution Phalloidin-488 (Thermo Fisher, A12379) and incubated in a humid chamber (room temperature, 1 hr). Triplicate washes of 0.1% Triton-X in 1% BSA/PBS were carried out (room temperature, 10 min) before mounting with aqueous mounting media (Abcam, ab128982). Immunostained sections were imaged with an inverted spinning disk confocal microscope (Nikon, NISElements Viewer 5.2 software). Optimal z-axis step size was determined according to the Nyquist criteria for sampling. Images were post-processed and analyzed through ImageJ/FIJI.

## Primary antibodies

Anti-Endomucin (Emcn) 1:100 (Santa Cruz, sc-65495)
Anti-CD90/Thy1, 1:100 (Abcam, ab3105)
Anti-phosphoAMPK (Thr183, Thr172), 1:100 (Thermo Fisher, 44–1150 G)
Anti-ATPB, 1:100 (Abcam, ab128743)
Anti-Ki67, 1:100 (Thermo Fisher, MA5-14520)
Anti-p16INK4a, 1:100 (Abcam, ab211542)
Anti-beta-catenin 1:100 (Abcam, ab32572)
Anti-MMP2, 1:100 (Abcam ab92536)
Anti-CD31, 1:100 (Novus Biologicals, NB100-2284)
Anti-Lamin B1, 1:100 (Abcam, ab16048)

## Secondary antibodies

Chicken anti-Rat-647 1:1000 (Thermo Fisher, A21472)
Donkey anti-Rabbit-555 1:1000 (Thermo Fisher, A32816)

## Fluorescence thresholding quantification

See *Figure 1—figure supplement 1*. Sample tissue sections were immunostained alongside controls that lacked primary antibody: the 'no primary control'. These control samples were imaged using the same microscope settings as the experimental samples, and maximum projections were calculated to reduce volumetric images to 2D. The maximum pixel intensity in the control images was used to threshold the experimental samples, thereby excluding signal arising from non-specific staining by fluorescent secondary antibodies. Cellular localization of the remaining signal allowed designation of cells positive and negative for the antibody targets of interest.

For quantitative measurements of F-actin/Phalloidin-488 signal intensity (e.g. *Figure 1h*), sum-projections were taken to reduce volumetric 3D images to 2D space. In these quantifications, mean sum pixel intensity was measured and the mean sum background pixel intensity subtracted for $N \geq 56$ cells in n=3 mice.

## Tetramethylrhodamine (TMRM) and Hoechst staining in live tissue ex vivo

Periosteum was dissected from n=3 freshly killed mice, and immediately placed in warmed culture media (10% FBS in DMEM) containing 100 nM TMRM (Thermo Fisher, I34361) and 5 µg/mL Hoechst 33342 (Thermo Fisher, H3570) for 30 min at 37°C. Tissue was briefly washed with fresh warmed media and imaged on a Zeiss widefield microscope (Zeiss Zen Blue Edition 2.5 software). Explanted cell controls were allowed to migrate from minced suture mesenchyme and attach for 24 hr before identical TMRM/Hoechst staining.

## MC3T3 cell culture

MC3T3 (gift from Sophie Verrier, AO foundation, Davos, Switzerland) cells were cultured in MEMα (Thermo Fisher, A1049001) 10% foetal bovine serum (FBS; Thermo Fisher A5256701), 1% penicillin-streptomycin (PS; Thermo Fisher, 15140–122). Cells were used at passages 13–20 and seeded at 25,000 cells / cm² for further experiments in glucose- and pyruvate-free DMEM (Thermo Fisher, 11966025),

10% FBS, 1% PS. All experiments were conducted in differentiation medium, consisting of glucose- and pyruvate-free DMEM 10% FBS, 1% PS, 50 μM ascorbic acid (Merck, 5960), and 2 mM glycero-phosphate (Merck, 50020). Alternate serum deprivation was performed by replacing the osteogenic media by serum-free osteogenic media every 1 hr for 6 hr. For oligomycin and NMN treatments, cells were treated by 0.1 μM oligomycin (Merck, 75351), with or without 1 mM β-nicotinamide mononucle-otide (NMN; APExBio, B7878), or with 0.1% DMSO (Merck, D2438), the vehicle of oligomycin. Cells were treated for 6 hr before switching to differentiation medium ('oligomycin' or 'DMSO' condition) or differentiation medium +1 mM NMN ('oligomycin +NMN' condition). The media were changed twice a day.

## Inhibition of ATP production in MC3T3 and measure of Reactive Oxygen Species (ROS) production

ATP level in cells was measured in 96 well plates after 6 hr of the treatment induction, using the CellTiter-Glo 2.0 Assay (Promega, G9241) according to manufacturer's instructions. ROS production was measured in live cells using the CellROX deep red staining kit (Thermo Fisher, C10422) using a spinning disk microscope (Nikon Ti2, Crest Optics X-Light V3), with the sum projection intensity normalized by the number of cells using Hoechst staining.

## MC3T3 count and mitochondrial activity

Cell count and mitochondrial activity were measured with Hoechst and TMRM staining respectively, using a spinning disk microscope and according to manufacturer's instructions. Z-stacks of 20 μm were acquired and the TMRM sum intensity was normalized to the cell number.

## Holographic imaging

Mitochondrial morphology was observed using holotomography imaging on live cells (Tomocube, HT-X1), with a reference refractive index set at 1.337. After reconstruction of the images, mitochondria were categorized in three types according to their shape. The proportion of each morphology type was determined with Tomo Analysis and Fiji software. MitotrackerGreen (Thermo Fisher, M7514) was used to stain mitochondria and confirm the mitochondrial morphology obtained by holotomography.

## RT-qPCR

RNA was extracted according to manufacturer's instructions (Marchery-Nagel, M36006), 12 hr after the treatment induction. Reverse transcription (Thermofisher, 4374966) and qPCR (Thermofisher, 4368706) were performed according to manufacturer's instruction and mRNA target levels were normalized by beta-Actin mRNA levels using the $2^{-\Delta\Delta Ct}$ method. The following primers have been used (5'→ 3'): *Mfn1*: F-AACTTGATCGAATAGCATCCGAG, R-GCATTGCATTGATGACAGAGC; *Mfn2*: F-CTGGGGACCGGATCTTCTTC, R-CTGCCTCTCGAAATTCTGAAACT; *Drp1*: F-GGGCACTTAAAT TGGGCTCC, R-TGTATTCTGTTGGCGTGGAAC; *Pgc1a*: F-AGTGGTGTAGCGACCAATCG, R-AATG AGGGCAATCCGTCTTCA; *Arbp*: F-AGATTCGGGGATATGCTGTTGG, R-AAAGCCTGGAAGAAGG AGGTC; *Pink1*: F-CACACTGTTCCTCGTTATGAAGA, R- TTGAGATCCCGATGGGCAAT; *Parkin*: F-GAGGTCCAGCAGTTAAACCCA; R-CACACTGAACTCGGAGCTTTC; *Lc3II* F-TTATAGAGCGATACAA GGGGGAG, R-CGCCGTCTGATTATCTTGATGAG; *Atg5*: F-AAGTCTGTCCTTCCGCAGTC, R-TGAA GAAAGTTATCTGGGTAGCTCA; *Lonp1*: F-ATGACCGTCCCGGATGTGT, R-CCTCCACGATCTTGAT AAAGCG; *Hsp60*: F-CACAGTCCTTCGCCAGATGAG, R-CTACACCTTGAAGCATTAAGGCT; *Hpsa9*: F-AATGAGAGCGCTCCTTGCTG, R-CTGTTCCCCAGTGCCAGAAC; *Mtco1*: F-TATGTTCTATCAATGG GAGC, R-GTAGTCTGAGTAGCGTCGTG; *β-Actin*: F- GGCTGTATTCCCCTCCATCG, R- CCAGTTGG TAACAATGCCATGT.

## Evaluation of the osteoblastic differentiation

Cells were cultured for 2 weeks before the staining of the alkaline phosphatase (ALP; Abcam, ab284936) or mineralized matrix using alizarin red S (ARS; 2% solution in $H_2O$) (Merck, A5533). ALP staining was performed as per manufacturer's recommendation, and the stained area were quantified using Fiji on pictures taken under a Evos XL core microscope. For ARS staining the cells were fixed with 4% paraformaldehyde in PBS, stained by ARS then imaged using a Evos XL core microscope. The

stained mineralized matrix was then dissolved with 10% (v/v) acetic acid in $H_2O$ and optical density was measured at 405 nm to quantify the amount of dissolved ARS.

## Statistical analysis

Statistical analysis was performed in GraphPad Prism 9. All data are plotted as mean, with error bars indicating standard deviation. Typically, experimental data was analyzed from all mouse conditions together using one-way ANOVA, with individual comparisons between conditions/regions of interest displayed on figures as Welch's two-tailed unpaired tests. Welch's ANOVA was used instead of ordinary ANOVA because it is more conservative and reduces the risk of false-positive p-values. Two-factors comparisons were analyzed with two-way ANOVA with uncorrected Fisher's LSD test. Only the statistical values for the hypotheses being tested are shown; multiple comparison corrections were not applied. Ratio-paired two-tailed t-test was used in comparisons within the same mouse (i.e. comparisons of Wnt3a-bandage vs Inactive-bandage). Correction of multiple comparison was not applied as each p-value was considered for itself. On some tested groups, correction for multiple comparison did not change our general conclusions. Uncorrected Dunn's tests were performed for the statistical analysis of UCell gene-expression signature scoring, as a non-parametric test for non-Gaussian distributions of data.

## RNAseq sample processing

Periosteum and suture mesenchyme were separately dissected and pooled from ≥3 live mice from each condition, briefly minced with a scalpel blade, and immediately placed in 250 µL of a solution of 0.5 mg/ml Liberase TL (Roche, 05401020001) and 0.1 mg/ml DNaseI (Sigma-Aldrich, DN25) in Hanks' Balanced Salt Solution (HBSS, Gibco, 14170–120) and incubated for 25 min at 37°C with shaking (600 rpm). The digestion was quenched with ice-cold 1 mL buffer (3% FBS in PBS), passed through a 100 µm cell strainer (Corning, 352360), and centrifuged (500 × $g$, 5 min, 4°C). The supernatant was removed, and the cell pellet gently resuspended in staining solution containing 0.1 µg 'hashtag' antibody, 50 µM DRAQ5 (Invitrogen, 65-0880-92), and 1:4000 DAPI in ice-cold 3% FBS/PBS. Up to 4 samples were individually stained with TotalSeq antibodies 1–4 (Biolegend 155801, 155803, 155805, 155807). Samples were centrifuged, washed with 3% FBS/PBS, centrifuged again and resuspended in 3% FBS/PBS for single-cell FACS. The gating strategy took single (by forward and side scatter), nucleated (DRAQ⁺), live (DAPI⁻) cells, and up to 4 individually-hashtagged samples were multiplexed together by sorting equal cell numbers into the same collection tube, and stored on ice to preserve viability. Cells were loaded onto a 10 X Chromium controllers and the library generated through single cell 10X3' v3.1 gene expression. All libraries were sequenced together using NextSeq 2000 sequencing on a P3 flow cell.

## RNAseq data processing

Initial alignment and processing were performed through the CellRanger pipeline. Further quality control, processing, and analysis was carried out through the Seurat (v4) package (*Hao et al., 2021*) and additional packages in R software (*R Development Core Team, 2022*, v4.2.1). Cells were graded for proportion of mitochondrial genes, and this metric regressed. The immune cell CD45⁺ populations in the samples (*Ptprc*ʰⁱᵍʰ) were removed from downstream analyses. CD90⁺/CD45⁻ (*Thy1*ʰⁱᵍʰ/*Ptprc*ˡᵒʷ) clusters were considered the populations of interest. Differential expression of lincRNA Gm42418 was identified as an artefact and removed from the data, as has been done in other studies (*Kimmel et al., 2019*). For calculation of Gene Ontology (GO) and KEGG enrichment of sets of differentially expressed genes, ShinyGO 0.76.3 (*Ge et al., 2020*) was used to calculate false discovery rate (FDR) and fold enrichment (FE) values. Full details of differentially expressed genes in comparisons between ages, diets, and CD90⁺ subpopulations are available in *Source data 2*.

To construct the senescence signature, we obtained mouse aging-related genes from the GenAge database (https://genomics.senescence.info/genes/search.php?organism=Mus+musculus&show=4). The genes were categorized into two groups: positively regulated and negatively regulated. Positively regulated genes are defined as those with high expression levels associated with senescence, while negatively regulated genes are defined as those with low expression levels associated with senescence. The genes were clustered based on their expression, then we selected the highly expressed

cluster in the positively-regulated genes and the poorly expressed cluster in negatively-regulated genes in the Aged group.

For the glycolysis and lipid catabolic processes signatures, we obtained mouse metabolism-related genes from AmiGO2 database (https://amigo.geneontology.org/amigo) with Ontology ID: GO:0061621 (glycolysis) and GO:0050996 (positive regulation of lipid catabolic process), and we clustered the genes based on their expression levels. Heatmaps were drawn by R package pheatmap (version 1.0.12).

## Acknowledgements

We thank Dr Nadège Zanou for discussions about the oligomycin experiments. We also thank Prof Mehdi Tafti and Dr Frédéric Schütz for discussions on statistics. RC salary is supported by MRC MR/S023747/1. This research was supported by funds from the University of Lausanne, Switzerland (SJH). Additional financial support from the UK Regenerative Medicine Platform (MR/R015635/1, SJH).

## Additional information

### Competing interests

Shukry J Habib: Listed as an inventor on the patent application WO2021156519A1: Tissue regeneration patch. The other authors declare that no competing interests exist.

### Funding

| Funder | Grant reference number | Author |
| --- | --- | --- |
| University of Lausanne | | Joshua Reeves<br>Pierre Tournier<br>Pierre Becquart<br>Shukry J Habib |
| Medical Research Council | MR/R015635/1 | Joshua Reeves<br>Shukry J Habib |
| Medical Research Council | MR/S023747/1 | Robert Carton |
| UK Regenerative Medicine Platform | MR/R015635/1 | Shukry J Habib |

The funders had no role in study design, data collection and interpretation, or the decision to submit the work for publication.

### Author contributions

Joshua Reeves, Data curation, Formal analysis, Validation, Investigation, Visualization, Methodology, Writing – original draft, Writing – review and editing; Pierre Tournier, Pierre Becquart, Data curation, Formal analysis, Validation, Investigation, Visualization, Methodology, Writing – review and editing; Robert Carton, Yin Tang, Formal analysis, Methodology, Writing – review and editing; Alessandra Vigilante, Dong Fang, Formal analysis, Supervision, Validation, Methodology, Writing – review and editing; Shukry J Habib, Conceptualization, Resources, Formal analysis, Supervision, Funding acquisition, Visualization, Methodology, Writing – original draft, Project administration, Writing – review and editing

### Author ORCIDs

Pierre Tournier ⓘ https://orcid.org/0000-0001-7593-6928
Pierre Becquart ⓘ https://orcid.org/0000-0003-1462-0406
Alessandra Vigilante ⓘ https://orcid.org/0000-0002-8768-8287
Shukry J Habib ⓘ https://orcid.org/0000-0003-3132-2216

### Ethics

All procedures involving animals were carried out by holders of Personal Licences (PIL) under a Project Licence (PPL, P8F273FDD, S.J.H.) shaped by ARRIVE and NC3Rs guidelines. The PPL was approved by the Animal Welfare and Ethical Review Body (AWERB) at King's College London and granted by the

Home Office under the Animals (Scientific) Procedures Act 1986 (ASPA) and Amendment Regulations 2012.

Reviewer #1 (Public review): https://doi.org/10.7554/eLife.104068.3.sa1
Reviewer #2 (Public review): https://doi.org/10.7554/eLife.104068.3.sa2
Reviewer #3 (Public review): https://doi.org/10.7554/eLife.104068.3.sa3
Author response https://doi.org/10.7554/eLife.104068.3.sa4

## Additional files

### Supplementary files
• Supplementary file 1. Detailed statistical analysis of *Figure 1o*. Stars and symbols indicate statistical significance calculated by one-way ANOVA followed by unpaired t-test with Welch's correction, or by ratio paired t-test (non-significant, *P*>0.05) when comparing inactive vs active within the same group (e.g. Young inactive vs Young active). Ns: non significant, *\*P*<0.05, *\*\*P*<0.01, *\*\*\*P*<0.001, *\*\*\*\*P*<0.0001.

• Supplementary file 2. RNAseq comparison data summary.

• MDAR checklist

• Source code 1. R source code used to replicate the processed ".rds" file from the raw RNAseq data.

• Source code 2. R source code used to generate graphs for *Figure 2* and associated Figure supplements and *Figure 4* Detailed statistical results and numerical values displayed are available in *Source data 2*.

• Source code 3. R source code used to generate graphs for *Figure 2—figure supplement 4*. Detailed statistical results and numerical values displayed are available in *Source data 2*.

• Source code 4. R source code used for the UCell processing for gene signature scoring. Detailed statistical results and numerical values displayed are available in *Source data 2*.

• Source data 1. Source data from the main figures.

• Source data 2. Detailed statistical results and source data of the RNAseq.

• Source data 3. Source Data for the figure supplements.

### Data availability
Sequencing data have been deposited in GEO under accession codes GSE226681. All data generated or analysed during this study are included in the manuscript and supporting files; source data files have been provided.

The following dataset was generated:

| Author(s) | Year | Dataset title | Dataset URL | Database and Identifier |
|---|---|---|---|---|
| Reeves J, Tournier P, Becquart P, Carton R, Tang Y, Vigilante A, Fang D, Habib SJ | 2024 | Rejuvenating the aging skeletal stem cell compartment for bone repair | https://www.ncbi.nlm.nih.gov/geo/query/acc.cgi?acc=GSE226681 | NCBI Gene Expression Omnibus, GSE226681 |

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

# Appendix 1

**Appendix 1—key resources table**

| Reagent type (species) or resource | Designation | Source or reference | Identifiers | Additional information |
|---|---|---|---|---|
| Cell line (*Mus musculus*) | MC3T3 | Cell line obtained from Sophie Verrier (AO foundation, Davos Switzerland) | CVCL_0D74 | |
| Cell line (*Mus musculus*) | Comma D beta | *Okuchi et al., 2021* | CVCL_5733 | |
| Cell line (*Mus musculus*) | LS/L | *Lowndes et al., 2016* | | |
| Antibody | Anti-Endomucin (Emcn) Rat monoclonal | Santa Cruz | sc-65495 | 1:100 |
| Antibody | Anti-CD90/Thy1 Rat monoclonal | Abcam | ab3105 | 1:100 |
| Antibody | Anti-phosphoAMPK Rabbit polyclonal | Thermo Fisher | 44–1150 G | 1:100 |
| Antibody | Anti-ATPB Rabbit polyclonal | Abcam | ab128743 | 1:100 |
| Antibody | Anti-Ki67 Rabbit monoclonal | Thermo Fisher | MA5-14520 | 1:100 |
| Antibody | Anti-p16INK4a Rabbit monoclonal | Abcam | ab211542 | 1:100 |
| Antibody | Anti-beta-catenin Rabbit monoclonal | Abcam | ab32572 | 1:100 |
| Antibody | Anti-MMP2 Rabbit monoclonal | Abcam | ab92536 | 1:100 |
| Antibody | Anti-CD31Rabbit polyclonal | Novus Biologicals | NB100-2284 | 1:100 |
| Antibody | Chicken anti-Rat-647 | Thermo Fisher | A21472 | 1:1000 |
| Antibody | Donkey anti-Rabbit-555 | Thermo Fisher | A32816 | 1:1000 |
| Antibody | TotalSeq antibodies 1–4 | Biolegend | 155801, 155803, 155805, 155807 | |
| Recombinant protein (*Mus musculus*) | Wnt3a | R&D Systems | 1324-WN-010 | |
| Commercial Kit | CellTiter-Glo 2.0 Assay | Promega | G9241 | |
| Commercial Kit | CellROX deep red staining kit | Thermo Fisher | C10422 | |
| Commercial Kit | MitotrackerGreen | Thermofisher | M7514 | |
| Commercial Kit | RNA extraction kit | Marchery-Nagel | M36006 | |
| Commercial Kit | Reverse Transcription kit | Thermo Fisher | 4374966 | |
| Commercial Kit | qPCR Kit | Thermo Fisher | 4368706 | |
| Commercial Kit | ALP staining kit | Abcam | ab284936 | |
| Chemical compound, Drug | APTES | Sigma | 440140 | |
| Chemical compound, Drug | succinic anhydride | Sigma | 239690–50 G | |

*Appendix 1 Continued on next page*

*Appendix 1 Continued*

| Reagent type (species) or resource | Designation | Source or reference | Identifiers | Additional information |
|---|---|---|---|---|
| Chemical compound, Drug | MES | Sigma | M3671 | |
| Chemical compound, Drug | EDC | Sigma | E7750 | |
| Chemical compound, Drug | NHS | Sigma | 56480 | |
| Chemical compound, Drug | DTT | Roche | 10197777001 | |
| Chemical compound, Drug | NMN | APExBIO | B7878 | |
| Chemical compound, Drug | Glycerol | Sigma | G2025 | |
| Chemical compound, Drug | GAM-Agar | Sigma | 01916 | |
| Chemical compound, Drug | Tween-20 | Sigma | P9416 | |
| Chemical compound, Drug | Tris | Sigma | 93362 | |
| Chemical compound, Drug | EDTA | Sigma | E6758 | |
| Chemical compound, Drug | Triton X-100 | Sigma | 93443 | |
| Chemical compound, Drug | Bovine Serum Albumin | Sigma | A2153 | |
| Chemical compound, Drug | Phosphate Buffered Saline | Sigma | P4417 | |
| Chemical compound, Drug | DAPI | Sigma | D9542 | |
| Chemical compound, Drug | Phalloidin-488 | Thermo Fisher | A12379 | |
| Chemical compound, Drug | Aqueous mounting media | Abcam | ab128982 | |
| Chemical compound, Drug | TMRM | Thermo Fisher | I34361 | |
| Chemical compound, Drug | Hoechst | Thermo Fisher | H3570 | |
| Chemical compound, Drug | Ascorbic acid | Merck | 5960 | |
| Chemical compound, Drug | Glycerophosphate | Merck | 50020 | |
| Chemical compound, Drug | Oligomycin | Merck | 75351 | |
| Chemical compound, Drug | DMSO | Merck | D2438 | |
| Chemical compound, Drug | Alizarin Red S | Merck | A5533 | |
| Chemical compound, Drug | DRAQ5 | Invitrogen | 65-0880-92 | |

*Appendix 1 Continued on next page*

*Appendix 1 Continued*

| Reagent type (species) or resource | Designation | Source or reference | Identifiers | Additional information |
|---|---|---|---|---|
| Chemical compound, Drug | Liberase TL | Roche | 05401020001 | |
| Chemical compound, Drug | DNaseI | Sigma | DN25 | |
| Chemical compound, Drug | Collagen | Corning | 354249 | |
| Chemical compound, Drug | Surgical glue | 3 M, Vetbond | | |
| Strain (*Mus musculus*) | C57BL6/J | Charles River | Strain 632 | Females |
| Strain (Akkermansia muciniphila) | Akkermansia muciniphila | Institut Pasteur | CIP107961T | |
| Other | Fetal Bovine Serum | Sigma | F7524 | Cell culture media supplement |
| Other | DMEM | Thermo Fisher | 41966 | Cell culture media |
| Other | Gifu's Anaerobic Medium | Nissui Pharmaceutical | 05422 | Bacteria culture media |
| Other | MEM alpha | Thermo Fisher | A1049001 | Cell culture media |
| Other | Panicillin-Streptomycin | Thermo Fisher | 15140–122 | Antibiotics for culture media |
| Other | glucose- and pyruvate-free DMEM | Thermo Fisher | 11966025 | Cell culture media |
| Other | Polycaprolactone polymer film | Ploy-Med | | Bio-compatible material for Wnt-bandage |
| Other | HBSS | Gibco | 14170–120 | Buffered solution |
| Sequence-Based Reagent | MFN1 | Themo Fisher | F- ACTTGATCGAATAGCATCCGAG R-GCATTGCATTGATGACAGAGC | |
| Sequence-Based Reagent | MFN2 | Themo Fisher | F-CTGGGGACCGGATCTTCTTC R-CTGCCTCTCGAAATTCTGAAACT | |
| Sequence-Based Reagent | DRP1 | Themo Fisher | F-GGGCACTTAAATTGGGCTCC R-TGTATTCTGTTGGCGTGGAAC | |
| Sequence-Based Reagent | PGC1a | Themo Fisher | F-AGTGGTGTAGCGACCAATCGR-AATGAGGGCAATCCGTCTTCA | |
| Sequence-Based Reagent | ARBP | Themo Fisher | F-AGATTCGGGATATGCTGTTGG R-AAAGCCTGGAAGAAGGAGGTC | |
| Sequence-Based Reagent | PINK1 | Themo Fisher | F-CACACTGTTCCTCGTTATGAAGA R- TTGAGATCCCGATGGGCAAT | |
| Sequence-Based Reagent | PARKIN | Themo Fisher | F-GAGGTCCAGCAGTTAAACCCA R-CACACTGAACTCGGAGCTTTC | |
| Sequence-Based Reagent | LC3II | Themo Fisher | F-TTATAGAGCGATACAAGGGGGAG R-CGCCGTCTGATTATCTTGATGAG | |
| Sequence-Based Reagent | ATG5 | Themo Fisher | F-AAGTCTGTCCTTCCGCAGTC R-TGAAGAAAGTTATCTGGGTAGCTCA | |
| Sequence-Based Reagent | LONP1 | Themo Fisher | F-ATGACCGTCCCGGATGTGT R-CCTCCACGATCTTGATAAAGCG | |

*Appendix 1 Continued*

| Reagent type (species) or resource | Designation | Source or reference | Identifiers | Additional information |
|---|---|---|---|---|
| Sequence-Based Reagent | HSP60 | Themo Fisher | F-CACAGTCCTTCGCCAGATGAG R-CTACACCTTGAAGCATTAAGGCT | |
| Sequence-Based Reagent | HPSA9 | Themo Fisher | FAATGAGAGCGCTCCTTGCTG R-CTGTTCCCCAGTGCCAGAAC | |
| Sequence-Based Reagent | MTCO1 | Themo Fisher | F-TATGTTCTATCAATGGGAGCR-GTAGTCTGAGTAGCGTCGTG | |
| Software | SCANCO Visualizer 1.1 | Scanco Medical | | |
| Software | NISElements 5.2 | Nikon | | |
| Software | ImageJ | ImageJ | | |
| Software | Prism | GraphPad | | |
| Software | R | R | | |

