## [Editor Report · eLife Assessment]

Aging reduces tissue regeneration capacity, posing challenges for an aging population. In this **fundamental** study, Reeves et al. show that by combining Wnt-mediated osteoprogenitor expansion (using a special bandage) with intermittent fasting, calvarial bone healing can be restored in aged animals. Intermitted fasting improves osteoprogenitor function by rescuing aging-related mitochondrial dysfunction, which can also be achieved by nicotinamide mononucleotide (NMN) supplementation or by modulating the gut microbiome. By employing rigorous histological, transcriptomic, and imaging analyses in a clinically relevant model, the authors provide **compelling** evidence supporting the conclusions. The therapeutic approach presented in this study shows promise for rejuvenating tissue repair, not only in bones but potentially across other tissues.

---

## [Referee Report · Reviewer #1 (Public review)]

Aging reduces tissue regeneration capacity, posing challenges for an aging population. In this study, the authors investigate impaired bone healing in aging, focusing on calvarial bones, and introduce a two-part rejuvenation strategy. Aging depletes osteoprogenitor cells and reduces their function, which hinders bone repair. Simply increasing the number of these cells does not restore their regenerative capacity in aged mice, highlighting intrinsic cellular deficits. The authors' strategy combines Wnt-mediated osteoprogenitor expansion with intermittent fasting, which remarkably restores bone healing. Intermittent fasting enhances osteoprogenitor function by targeting NAD+ pathways and gut microbiota, addressing mitochondrial dysfunction - an essential factor in aging. This approach shows promise for rejuvenating tissue repair, not only in bones but potentially across other tissues.

This study is exciting, impressive, and novel. The data presented is robust and supports the findings well.

---

## [Referee Report · Reviewer #2 (Public review)]

Reeves et al explore a model of bone healing in the context of aging. They show that intermittent fasting can improve bone healing, even in aged animals. Their study combines a 'bone bandage' which delivers a canonical Wnt signal with intermittent fasting and shows impacts on the CD90 progenitor cell population and the healing of a critical-sized defect in the calvarium. They also explore potential regulators of this process and identify mitochondrial dysfunction in the age-related decline of stem cells. In this context, by modulating NAD+ pathways or the gut microbiota, they can also enhance healing, hinting at an effect mediated by complex impacts on multiple pathways associated with cellular metabolism.

The study shows a remarkable finding: that age-related decreases in bone healing can be restored by intermittent fasting. There is ample evidence that intermittent fasting can delay aging, but here the authors provide evidence that in an already-aged animal, intermittent fasting can restore healing to levels seen in younger animals. This is an important finding as it may hint at the potential benefits of intermittent fasting in tissue repair.

---

## [Referee Report · Reviewer #3 (Public review)]

Summary:

This study aims to address the significant challenge of age-related decline in bone healing by developing a dual therapeutic strategy that rejuvenates osteogenic function in aged calvarial bone tissue. Specifically, the authors investigate the efficacy of combining local Wnt3a-mediated osteoprogenitor stimulation with systemic intermittent fasting (IF) to restore bone repair capacity in aged mice. The highlights are:

(1) Novel Approach with Aged Models:

This pioneering study is among the first to demonstrate the rejuvenation of osteoblasts in significantly aged animals through intermitted fasting, showcasing a new avenue for regenerative therapies.

(2) Rejuvenation Potential in Aged Tissues:

The findings reveal that even aged tissues retain the capacity for rejuvenation, highlighting the potential for targeted interventions to restore youthful cellular function.

(3) Enhanced Vascular Health:

The study also shows that vascular structure and function can be significantly improved in aged tissues, further supporting tissue regeneration and overall health.

Through this innovative approach, the authors seek to overcome intrinsic cellular deficits and environmental changes within aged osteogenic compartments, ultimately achieving bone healing levels comparable to those seen in young mice.

Strengths:

The study is a strong example of translational research, employing robust methodologies across molecular, cellular, and tissue-level analyses. The authors leverage a clinically relevant, immunocompetent mouse model and apply advanced histological, transcriptomic, and functional assays to characterise age-related changes in bone structure and function. Major strengths include the use of single-cell RNA sequencing (scRNA-seq) to profile osteoprogenitor populations within the calvarial periosteum and suture mesenchyme, as well as quantitative assessments of mitochondrial health, vascular density, and osteogenic function. Another important point is the use of very old animals (up to 88 weeks, almost 2 years) modelling the human bone aging that usually starts >65 yo. This comprehensive approach enables the authors to identify critical age-related deficits in osteoprogenitor number, function, and microenvironment, thereby justifying the combined Wnt3a and IF intervention.

[Editors' note: The manuscript was evaluated positively by all three reviewers originally. In the revised manuscript, the authors included some new data following the reviewers' suggestions, while other comments were clarified in the response to the reviewers, and by revising the manuscript text. The new data further support the major conclusions of the paper.]

---

## [Author Response]

The following is the authors’ response to the original reviews.

**Public Reviews:**

**Reviewer #1 (Public review):**
Summary:Aging reduces tissue regeneration capacity, posing challenges for an aging population. In this study, the authors investigate impaired bone healing in aging, focusing on calvarial bones, and introduce a two-part rejuvenation strategy. Aging depletes osteoprogenitor cells and reduces their function, which hinders bone repair. Simply increasing the number of these cells does not restore their regenerative capacity in aged mice, highlighting intrinsic cellular deficits. The authors' strategy combines Wnt-mediated osteoprogenitor expansion with intermittent fasting, which remarkably restores bone healing. Intermittent fasting enhances osteoprogenitor function by targeting NAD+ pathways and gut microbiota, addressing mitochondrial dysfunction - an essential factor in aging. This approach shows promise for rejuvenating tissue repair, not only in bones but potentially across other tissues.Strengths:This study is exciting, impressive, and novel. The data presented is robust and supports the findings well.Weaknesses:As mentioned above the data is robust and supports the findings well. I have minor comments only.

We thank the reviewer for their enthusiastic and positive assessment of our study. We appreciate the recognition of the novelty and robustness of our data and findings. We have carefully considered the reviewer's comments and have revised the manuscript accordingly. We believe these revisions further strengthen the clarity and impact of our work.

**Reviewer #2 (Public review):**
Summary:Reeves et al explore a model of bone healing in the context of aging. They show that intermittent fasting can improve bone healing, even in aged animals. Their study combines a 'bone bandage' which delivers a canonical Wnt signal with intermittent fasting and shows impacts on the CD90 progenitor cell population and the healing of a critical-sized defect in the calvarium. They also explore potential regulators of this process and identify mitochondrial dysfunction in the age-related decline of stem cells. In this context, by modulating NAD+ pathways or the gut microbiota, they can also enhance healing, hinting at an effect mediated by complex impacts on multiple pathways associated with cellular metabolism.Strengths:The study shows a remarkable finding: that age-related decreases in bone healing can be restored by intermittent fasting. There is ample evidence that intermittent fasting can delay aging, but here the authors provide evidence that in an already-aged animal, intermittent fasting can restore healing to levels seen in younger animals. This is an important finding as it may hint at the potential benefits of intermittent fasting in tissue repair.Weaknesses:The authors explore potential mechanisms by which the intermittent fasting protocol might impact bone healing. However, they do not identify a magic bullet here that controls this effect. Indeed, the fact that their results with intermittent fasting can be replicated by changing the gut microbiota or modulating fundamental pathways associated with NAD, suggests that there is no single mechanism that drives this effect, but rather an overall complex impact on metabolic processes, which may be very difficult to untangle.

We thank the reviewer for their positive assessment of our study and for highlighting the significant finding that intermittent fasting can restore age-related declines in bone healing. We appreciate the observation that our results suggest a complex interplay of metabolic processes rather than a single "magic bullet" mechanism. Indeed, the ability of gut microbiota modulation or NAD+ pathway targeting to replicate intermittent fasting's benefits underscores this complexity. While we recognize the challenges of disentangling these interconnected pathways, we believe our findings offer valuable insights into the multifaceted nature of intermittent fasting's impact on aged tissue repair. We hope this study serves as a foundation for future research aimed at identifying the individual contributions of these pathways and developing targeted therapeutic strategies.

**Reviewer #3 (Public review):**
Summary:This study aims to address the significant challenge of age-related decline in bone healing by developing a dual therapeutic strategy that rejuvenates osteogenic function in aged calvarial bone tissue. Specifically, the authors investigate the efficacy of combining local Wnt3a-mediated osteoprogenitor stimulation with systemic intermittent fasting (IF) to restore bone repair capacity in aged mice. The highlights are:(1) Novel Approach with Aged Models:This pioneering study is among the first to demonstrate the rejuvenation of osteoblasts in significantly aged animals through intermitted fasting, showcasing a new avenue for regenerative therapies.(2) Rejuvenation Potential in Aged Tissues:The findings reveal that even aged tissues retain the capacity for rejuvenation, highlighting the potential for targeted interventions to restore youthful cellular function.(3) Enhanced Vascular Health:The study also shows that vascular structure and function can be significantly improved in aged tissues, further supporting tissue regeneration and overall health.Through this innovative approach, the authors seek to overcome intrinsic cellular deficits and environmental changes within aged osteogenic compartments, ultimately achieving bone healing levels comparable to those seen in young mice.Strengths:The study is a strong example of translational research, employing robust methodologies across molecular, cellular, and tissue-level analyses. The authors leverage a clinically relevant, immunocompetent mouse model and apply advanced histological, transcriptomic, and functional assays to characterise age-related changes in bone structure and function. Major strengths include the use of single-cell RNA sequencing (scRNA-seq) to profile osteoprogenitor populations within the calvarial periosteum and suture mesenchyme, as well as quantitative assessments of mitochondrial health, vascular density, and osteogenic function. Another important point is the use of very old animals (up to 88 weeks, almost 2 years) modelling the human bone aging that usually starts >65 yo. This comprehensive approach enables the authors to identify critical age-related deficits in osteoprogenitor number, function, and microenvironment, thereby justifying the combined Wnt3a and IF intervention.Weaknesses:One limitation is the use of female subjects only and the limited exploration of immune cell involvement in bone healing. Given the known role of the immune system in tissue repair, future studies including a deeper examination of immune cell dynamics within aged osteogenic compartments could provide further insights into the mechanisms of action of IF.

We thank the reviewer for their thorough summary and positive assessment of our study, particularly highlighting its translational nature, the robust methodologies employed, and the relevance of our aged animal model. We appreciate the insightful suggestion to include male subjects and to explore immune cell dynamics in future investigations.

We acknowledge the limitation of using only female mice in the current study and agree that future studies incorporating both sexes and investigating immune cell contributions within aged osteogenic compartments would offer valuable insights into the mechanisms underlying intermittent fasting and its impact on bone healing.

Our focus on female mice was informed by their distinct characteristics, including delayed healing and higher fracture risk (PMID: 37508423, PMID: 34434120). Importantly, female mice present a more challenging case for bone repair, making them a stringent test for evaluating the effectiveness of our rejuvenation approaches. Moreover, our research protocol, approved under animal license, adhered to ethical principles and the 3Rs, allowing us to reduce the number of animals required by focusing on a single sex.

**Recommendations for the authors:**

**Reviewer #1 (Recommendations for the authors):**
(1) The authors should provide a justification for the use of female mice in this study. Additionally, the section on animal methods should be expanded to align with ARRIVE guidelines.

We thank the reviewer for their valuable feedback. In response to the comment regarding the use of female mice, we have included a justification in the updated manuscript. As noted, female mice were selected for this study due to their distinct characteristics, such as delayed healing and higher fracture risk (PMID: 37508423, PMID: 34434120), which provide a more challenging model for evaluating bone repair strategies. We believe this made our study a stringent test of the efficacy of the rejuvenation approaches being investigated.

Additionally, we have revised the animal methods section to ensure it aligns with the ARRIVE guidelines.

(2) Intermittent fasting can influence circadian rhythms in various ways. In the RNA-seq data, do the authors observe any changes related to circadian rhythm pathways?

The reviewer raises an important point regarding the influence of intermittent fasting (IF) on circadian rhythms. Our RNA-seq data revealed significant alterations in circadian rhythm pathways, particularly within the aged periosteal CD90+ cell population during IF. Specifically, the PAR bZip family transcription factors Dbp, Hlf, and Tef (q < 0.05) were significantly upregulated, consistent with their established roles as circadian rhythm regulators (PMID: 16814730, PMID: 31428688).

In suture CD90+ cells from the Aged + IF group, Dbp expression was significantly elevated compared to the Aged AL control group. Moreover, several other circadian-controlled genes, including Sirt1, Kat2b, Csnk1e, Ezh2, Fbxw11, and Ucp2 (p < 0.05), were also upregulated (Fig. 4b), suggesting enrichment of Clock/Per2/Arntl transcriptional targets, essential components of the circadian clock.

The observed upregulation of circadian rhythm effectors like Dbp, Hlf, and Tef further suggests a potential role for circadian transcription in CD90+ cell rejuvenation and bone repair in aged mice. While previous studies have primarily focused on the role of circadian rhythms in osteoblasts in vitro (PMID: 34579752, PMID: 30290183), our findings provide compelling evidence for their involvement in bone regeneration in vivo, providing compelling evidence for future investigation into this mechanism.

Chip-SEQ studies have shown D-box sites near promoters in Wnt/β-catenin components (e.g. Lrp6, Lrp5, Wnt8a, Fzd4) in pro-osteogenic transcription factor Zbtb16 (and see Fig 5), and in 11 of the 44 mouse collagen genes (PMID: 31428688). These components are known to regulate osteogenesis, and their proximity to circadian-controlled transcription factors suggests a possible overlap between circadian regulation and Wnt signaling in promoting bone repair. Additionally, circadian rhythmicity, stem cell function, and Wnt signaling are interlinked (PMID: 29277155, PMID: 25414671). Food intake is a powerful regulator of the circadian rhythm in several organs (PMID: 11114885, PMID: 32363197), but little is known about the diet-circadian interaction in bone repair. The possibility that circadian transcription can be harnessed to target Aged stem cell function towards bone repair is a promising prospect.

We have incorporated this information in Figure 2 - figure supplement 3G-H, the results section as well as in the discussion.

**Reviewer #2 (Recommendations for the authors):**
(1) The authors refer to 'altered cellular mechanobiology', 'age-related changes in mechanobiology', etc. Here, they are using this terminology to refer to changes in F-actin intensity and nuclear shape. While I agree that these measures are indicators of a cellular response to mechanical cues, calling this 'changes in mechanobiology' doesn't sound quite correct to me. 'Mechanobiology' to me, is a field of study. Perhaps the authors should consider changing their terminology.

We appreciate the reviewer’s insightful comment on the terminology used in our manuscript. We agree that the term "mechanobiology" is a broad field of study and using it in the context of changes in F-actin intensity and nuclear shape may be misleading. We have revised the text to better reflect the specific cellular responses to mechanical cues, such as changes in the cytoskeleton and nuclear morphology, rather than referring to them as "altered mechanobiology." The updated terminology more accurately conveys the observed cellular alterations in response to mechanical forces. We have made these adjustments throughout the manuscript for clarity and precision.

(2) Three of the measures the authors use to highlight age-related changes (and rejuvenation) in their animal model are F-actin intensity, nuclear shape, and vascularisation. However, they never really explain what they believe these readouts mean practically/functionally. Indeed, it makes sense that less vascularisation would be associated with an aged phenotype and preclude healing, but this is only mentioned somewhat cursorily in the discussion. While vascularisation is discussed in the context of aging in the discussion, it is not discussed in the context of healing (which would seem relevant in the context of vascularisation being used as a readout in the healing models in response to Akk and IF treatment). Similarly, the changes in F-actin intensity and nuclear shape might suggest changes in the stiffness of the periosteum (as mentioned in the discussion), which could indeed be an indicator of an aged phenotype; however, their role in healing (in response to Akk and IF) are not clearly articulated.

We appreciate the reviewer’s insightful comments and have made revisions to clarify the implications of age-related changes in vascularization, F-actin intensity, and nuclear shape, as well as the functional significance of these observations in the context of healing and rejuvenation.

Vascularization:

Vascularization and modulation of blood flow are critical for calvarial bone repair, as supported by multiple studies (e.g., PMID: 38032405, PMID: 21156316, PMID: 25640220). Early in the calvarial repair process, blood vessels grow independently of osteoprogenitor cells, establishing a supportive environment that promotes osteoprogenitor migration and subsequent ossification (PMID: 38834586). Furthermore, angiogenic vessels from the periosteum at defect edges contribute to creating a specialized microenvironment essential for bone healing (PMID: 38834586, PMID: 38032405). Compromised vascularization significantly impairs the healing of critical-sized calvarial defects (PMID: 29702250).

Our data reveal a decline in periosteal vascularization with age, potentially compromising this microenvironment and impairing repair in aged animals. Importantly, our findings indicate that intermittent fasting (IF) reverses this phenotype by restoring periosteal vascularization. This rejuvenation of the vascular microenvironment aligns with improved bone repair outcomes in aged mice subjected to IF. We have revised the manuscript to emphasize the importance of vascularization in healing and to highlight the role of IF in restoring this critical aspect of the bone healing microenvironment.

F-actin intensity and nuclear shape:

Age-related changes in F-actin intensity and nuclear shape are associated with increased tissue stiffness, a hallmark of aging. Tissue stiffness has been shown to impair progenitor cell function and hinder repair in various systems, including neuroprogenitors (PMID: 31413369). Softening the extra cellular matrix in aged tissues has been demonstrated to partially restore progenitor function and improve repair outcomes, as seen in the case of neuroprogenitors (PMID: 31413369). In our study, IF reversed age-associated changes in F-actin expression and nuclear shape, restoring these parameters to a phenotype resembling that of younger animals. This suggests that IF mitigates the mechanical changes associated with aging, reducing tissue stiffness and rejuvenating the periosteum to facilitate improved bone healing, similar to the outcomes observed in younger models.

Following the reviewer’s advice, we have revised the text to clearly articulate the correlations and interpretations of our data regarding tissue mechanics and bone repair. Thank you for highlighting these critical aspects.

(3) In relation to my point (2) on nuclear shape, there are reports that aging is linked to changes in Lamin B1. Have the authors considered this? It might provide a clearer link between their data and the tissue-level phenotypes they observe.

Thank you for your comment regarding the potential link between aging and changes in Lamin B1. Following your suggestion, we performed Lamin B1 immunostaining on samples from Young, Adult, Aged, and Aged + IF groups. However, no significant differences in Lamin B1 levels were observed across these groups. These findings indicate that changes in Lamin B1 in osteoprogenitors are not apparent during aging, suggesting that Lamin B1 alterations in the context of aging may be tissue- and cell-type-specific.

The new data was added in Figure 1 - figure supplement 2i-j.

(4) In the data associated with Figure 2, the authors find that in the aged mice, MMP9 expression is increased, but MMP2 expression is decreased. They associate the decrease in MMP2 expression with decreased migration, but the canonical function of MMP9 should be similar to that of MMP2. Are there tissue-specific differences in the activity of MMP2/9 that could account for this?

Thank you for the thoughtful comment. While both MMP-2 and MMP-9 are involved in ECM remodeling and share some overlapping canonical functions, their roles are context-dependent and exhibit tissue-specific differences that could explain the observed changes in aged mice. MMP-2 has been shown to play a critical role in maintaining the structural and functional integrity of flat bones, such as those in the craniofacial skeleton, by supporting bone remodeling (PMID: 17400654, PMID: 17440987, PMID: 16959767). The decreased expression of MMP-2 in aged mice may impair these local processes, leading to reduced migratory capacity of osteoprogenitors and contributing to aging-related changes in flat bone structure and function.

In contrast, MMP-9 is more prominently involved in long bone remodeling, particularly at the growth plate where it regulates hypertrophic chondrocyte turnover, vascularization, and ossification during endochondral bone formation (PMID: 21611966, PMID: 9590175, PMID: 23782745, PMID: 16169742). Additionally, MMP-2 and MMP-9 differ in their regulation of specific ECM substrates and their interactions with bone-resident cells, which may further drive divergent outcomes in distinct bone types. For example, MMP-9’s role in osteoclastogenesis and its regulation of ECM proteins like type I collagen could be more critical in long bones, while MMP-2’s involvement in fine-tuning ECM microarchitecture may hold greater importance in flat bones.

The increased expression of MMP-9 in aged calvarial osteoprogenitors may reflect a compensatory mechanism in response to the reduced MMP-2 activity, possibly in response to increased ECM turnover demands. Further studies examining the precise molecular pathways driving these changes in osteoprogenitors will help clarify the underlying mechanisms and their contributions to age-related alterations in flat bone structure and function.

(5) In lines 391-2, the authors conclude that the data from Figure 4 shows that "during IF, CD90 cells, despite being aged, are more capable of ECM modulation and migration". The authors certainly present evidence that this is true, but the RNAseq showed that the enriched GO terms were predominantly associated with immune responses ('response to cytokine') and the proliferation phenotype seems very strong. Therefore, I would suggest that this overarching statement regarding the findings be less focussed on this one aspect of the finding, which doesn't look to be the dominant phenotype of the cellular response. And indeed, the authors move on from here to explore a mechanism associated with metabolism, not specifically with ECM remodelling.

We greatly appreciate the reviewer insight regarding the interpretation of our findings, particularly the conclusion drawn from Figure 4.

In response, we have revised the conclusion to more accurately reflect these findings.

The revised text in the conclusion now reads: " Together, these findings suggest that IF rejuvenates aged CD90+ cells, in part, by enhancing proliferation, immune response, ECM remodeling, Wnt/β-catenin pathway, and metabolism, including increased ATP levels and decreased AMPK levels.”

We hope that this adjustment better aligns with your suggestion and provides a more accurate summary of the key findings.

(6) Fasting blood glucose levels are often cited as an indicator of metabolic health. Did the authors look at this in their animals who underwent the IF protocol? Could this have had an impact on the healing response?

We thank the reviewer for this insightful comment. Throughout our study, we have withdrawn blood from the animals for various analyses that were not included in this manuscript in order to maintain focus on the osteoprogenitors.

Our analysis included the assessment of the metabolic health of the animals using fasting blood glucose levels and the area under the curve (AUC) of the intraperitoneal glucose tolerance test (IPGTT).

Fasting blood glucose levels reflect the animals' ability to maintain stable glucose levels after fasting, while the AUC from the IPGTT measures how efficiently glucose is cleared from the bloodstream following a glucose challenge. Typically, lower fasting blood glucose levels and reduced AUC indicate improved insulin sensitivity, better glucose metabolism, and enhanced metabolic control (PMID: 18812462, PMID: 19638507).

Our findings show that intermittent fasting (IF) significantly reduced both the fasting blood glucose levels and the AUC in the IPGTT. This indicates that IF enhances metabolic flexibility, likely through improved insulin sensitivity and better glucose homeostasis. By lowering fasting blood glucose, IF reduces the reliance on excessive gluconeogenesis during fasting, while a reduced AUC indicates more efficient postprandial glucose clearance, consistent with enhanced insulin action and reduced fluctuations in blood glucose levels. The new data has been incorporated in Figure 3 - figure supplement 1d-g.

Methods:

“Blood glucose level measurement

Fasting blood glucose levels were measured (Accu-Check tests strips) from 6h fasting mice by blood sampling the tail vein. For intraperitoneal glucose tolerance test (IPGTT), glucose was injected intraperitoneally (2 g/kg), and the blood glucose levels were measured after 15, 30, 60 and 120 minutes.”

Improved metabolic health through lower fasting glucose and reduced AUC can have profound implications for tissue repair (PMID: 32809434). Stable glucose levels ensure a consistent energy supply for key cellular processes, such as cell proliferation, migration, and differentiation, which are essential for regeneration. Enhanced insulin sensitivity supports nutrient delivery to cells and reduces inflammation, creating an environment conducive to tissue healing. Additionally, intermittent fasting's ability to optimize glucose metabolism and regulate insulin secretion may enhance the function of stem and progenitor cells, further improving the tissue repair process (PMID: 28843700). Together, these findings suggest a mechanistic link between improved metabolic health and the enhanced healing observed in animals subjected to intermittent fasting.

(7) In Supplementary Figure 10, the authors look at bone remodelling by assessing TRAP staining, as an indicator of osteoclast activity. I'm not sure if these data add all that much to the study. The authors have looked at bone formation at a tissue level using microCT. Here, they look at bone resorption at a cellular level with the TRAP assay. Overall, this probably suggests more bone remodelling, but the TRAP assay on its own at the cellular level could also be interpreted as an osteoporosis-like phenotype. This is clearly not the case because the authors show robust bone healing by microCT. In short, as an isolated measure of osteoclast activity at the cellular level without cellular-level assays of osteoblast activity, the interpretation of these data is not that clear. The microCT speaks far more of the phenotype and is, in my opinion, sufficient to make this point.

We thank the reviewer for their comments regarding the interpretation of the TRAP staining data and its context within the study. We appreciate the concern that, without direct assays of osteoblast activity, the TRAP assay could lead to ambiguity.

We have shown that intermittent fasting significantly increases the number and function of osteoprogenitor cells, the precursors to osteoblasts. While we acknowledge that these data do not directly measure osteoblast numbers or activity, they strongly suggest an increased capacity for osteoblast differentiation and bone formation. This aligns with the microCT findings of robust bone structure and healing.

After careful consideration and given that the microCT and histology findings already provide robust and comprehensive evidence for bone structure and healing, we have decided to remove the TRAP staining data from the manuscript. We believe this change simplifies the manuscript and strengthens its focus on the most impactful data.

(8) In the discussion, the authors make a number of links between aging and IF. However, one of the exciting conclusions of this manuscript is that IF aids in healing in aged animals. In this context, IF has not impacted the aging process itself because the animals have not experienced an IF protocol across their lifespan, but rather only after injury. In this context, perhaps the authors should also be focussing their discussion on evidence of the short-term response to IF rather than its effects on aging, which are longer-term.

We appreciate the reviewer's comment and agree that emphasizing the short-term effects of intermittent fasting is crucial. Our study is the first to examine this protocol in Aged animals.

To address this, we have revised the discussion and highlighted how short-term IF enhances metabolic health, promotes osteoprogenitor functionality, and supports bone remodeling, as observed in our study.

**Reviewer #3 (Recommendations for the authors):**
(1) The authors should clarify details on intermittent fasting protocols, especially regarding caloric intake differences between fasting and non-fasting days, to aid reproducibility.

We appreciate the reviewer's suggestions and have incorporated them by clarifying the relevant details. The new data are presented in Figure 3 - figure supplement 1a-c.

Methods:

“Caloric intake calculation

To assess the caloric intake of mice, the food was weighted when made available to the mice (Win), and when removed (Wout). The daily consumed food was calculated based on the weight difference (Win - Wout), then converted to kcal (1 g = 3.02 kcal, LabDiet, 5053), and expressed as kcal/mouse/day for each cage (n cage ³ 3 with 1 to 5 mice/cage).”

(2) Did the authors evaluate the effect of their intermittent fasting protocol on fasting blood glucose levels?

Following the reviewer comment we included two measurements: (1) Fasting blood glucose, which reflects the ability to maintain glucose homeostasis during fasting, and (2) fasting blood glucose levels and the area under the curve (AUC) of the intraperitoneal glucose tolerance test (IPGTT), which measures glucose clearance efficiency after a glucose challenge. Lower values for both typically indicate improved insulin sensitivity, glucose metabolism, and metabolic control.

Our findings demonstrate that intermittent fasting significantly reduced both fasting blood glucose and IPGTT AUC, suggesting enhanced metabolic flexibility, likely through improved insulin sensitivity and glucose homeostasis. Lower fasting blood glucose with IF indicates reduced reliance on gluconeogenesis during fasting, while a reduced AUC suggests more efficient postprandial glucose clearance, consistent with enhanced insulin action and reduced blood glucose fluctuations. This new data is included in Figure 3 - figure supplement 1.

Generally, the improved metabolic environment supports tissue repair by ensuring adequate energy for cell proliferation and migration, reducing inflammation, and promoting the function of stem cells involved in tissue regeneration. Thus, this outcome of intermittent fasting may create a more favorable environment for tissue repair, potentially accelerating the healing of damaged tissues and improving overall regenerative capacity.

(3) In Figure 1E-F, the nuclei have an interesting shape and the authors quantified F-actin. Given the role of lamin B in nuclear integrity, an analysis of lamin B expression and its structural integrity in aged osteoprogenitors could provide valuable insights into cellular aging mechanisms and their potential reversal with intermittent fasting.

In response to the reviewer's comment, we performed Lamin B1 immunostaining on samples from Young, Adult, Aged, and Aged + IF groups. We observed no significant differences in Lamin B1 levels across these groups. This suggests that age-related changes in Lamin B1 are not evident in osteoprogenitors and may be tissue- or cell-type specific. The new data was added in Figure 1 - figure supplement 2i-j.

(4) The authors should explain, in the main text or the methods section, why are they only using females in this study.

We appreciate the reviewer's comment regarding the use of female mice. Female mice were chosen for this study due to their delayed healing and higher fracture risk (PMID: 37508423, PMID: 34434120), presenting a more challenging model for evaluating bone repair strategies and providing a stringent test of our rejuvenation approaches. This justification has been added to the revised manuscript. The animal methods section has also been updated to comply with ARRIVE guidelines.

(5) This story stands alone and has an incredible amount of data. However, for a follow-up study, I would like to suggest consideration of including a broader analysis of immune cell involvement within the osteogenic compartments to strengthen the mechanistic understanding of IF's impact.

We thank the reviewer for this insightful suggestion. We agree that investigating the role of immune cells within the osteogenic compartments could provide valuable mechanistic insights into how intermittent fasting influences tissue regeneration. Immune cells are key mediators of inflammation and repair, and their interactions with osteoprogenitors and other cells in the bone healing environment likely contribute to IF's effects.

While our study focuses on IF's impact on osteoprogenitor function and tissue repair, we acknowledge the importance of future research exploring immune cell involvement. Techniques like single-cell RNA sequencing or flow cytometry could characterize immune cell populations and their functional states within osteogenic niches, allowing for a deeper understanding of immune-skeletal interactions during IF-mediated bone healing. We appreciate the reviewer highlighting this promising avenue for future research.

Minor corrections to the text and figures:(1) References formatting should be revised (eg. line 41).

The reference formatting was corrected.

(2) Line 144 - what do the authors mean by p2 in the references?

Thank you for your comment, we corrected the error and removed p2 from the reference.